# A Probabilistic Framework for Modular Continual Learning

## Abstract

Continual learning (CL) algorithms seek to accumulate and transfer knowledge across a sequence of tasks and achieve better performance on each successive task. Modular approaches, which use a different composition of modules for each task and avoid forgetting by design, have been shown to be a promising direction to CL. However, searching through the large space of possible module compositions remains a challenge. In this work, we develop a scalable probabilistic search framework as a solution to this challenge. Our framework has two distinct components. The first is designed to transfer knowledge across similar input domains. To this end, it models each module's training input distribution and uses a Bayesian model to find the most promising module compositions for a new task. The second component targets transfer across tasks with disparate input distributions or different input spaces and uses Bayesian optimisation to explore the space of module compositions. We show that these two methods can be easily combined and evaluate the resulting approach on two benchmark suites designed to capture different desiderata of CL techniques. The experiments show that our framework offers superior performance compared to state-of-the-art CL baselines.

## 1 Introduction

The *continual learning* (CL) (Thrun & Mitchell, 1995) setting calls for algorithms that can solve a sequence of learning problems while performing better on every successive problem. A CL algorithm should avoid catastrophic forgetting — i.e., not allow later tasks to overwrite what has been learned from earlier tasks — and achieve transfer across a large sequence of problems. Ideally, the algorithm should be able to transfer knowledge across similar input distributions (*perceptual transfer*), dissimilar input distributions and different input spaces (*non-perceptual transfer*), and to problems with a few training examples (*few-shot transfer*). It is also important that the algorithm's computational and memory demands scale sub-linearly with the number of encountered tasks.

Recent work (Valkov et al., 2018; Veniat et al., 2020; Ostapenko et al., 2021) has shown modular algorithms to be a promising approach to CL. These methods represent a neural network as a composition of modules, in which each module is a reusable parameterised function trained to perform an atomic transformation of its input. During learning, the algorithms accumulate a library of diverse modules by solving the encountered problems in a sequence. Given a new problem, they seek to find the best composition of pre-trained and new modules, out of the set of all possible compositions, as measured by the performance on a held-out dataset. Unlike CL approaches which share the same parameters across all problems, modular algorithms can introduce new modules and, thus, do not have an upper bound on the number of solved problems.

However, *scalability* remains a key challenge in modular approaches to CL, as the set of module compositions is discrete and explodes combinatorially. Prior work has often sidestepped this challenge by introducing various restrictions on the compositions, for example, by only handling perceptual transfer (Veniat et al., 2020) or by ignoring non-perceptual transfer and being limited by the number of modules that can be stored in memory (Ostapenko et al., 2021). The design of CL algorithms that relax these restrictions and can also scale remains an open problem.

In this paper, we present a probabilistic framework as a solution to the scalability challenges in modular CL. We observe that searching over module compositions efficiently is difficult because

Past solutions :

Set of all paths, $\Pi_4$ :

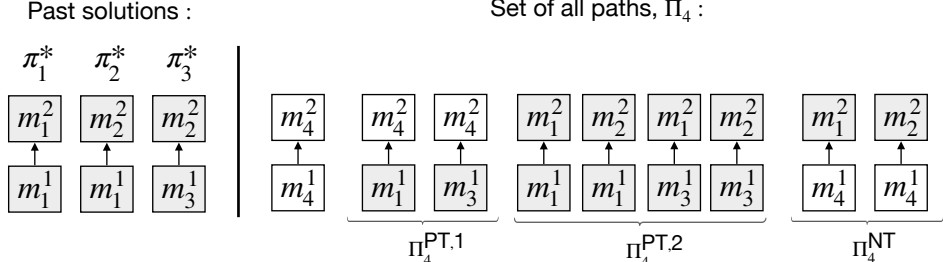

Figure 1: The figure depicts the set of all paths, $\Pi_4$, that a modular algorithm considers when solving the 4th problem $\Psi_4$ in a sequence. The modular architecture has $L = 2$ layers. The library comprises all previously trained modules: $\Lambda_4 = \{m_1^1, m_3^1, m_1^2, m_2^2\}$. Paths in $\Pi_4^{\text{PT},1}$ (Eq. 2) select a pre-trained module for the first layer, enabling perceptual transfer. Paths in $\Pi_4^{\text{PT},2}$ reuse modules in both layers. They can perform few-shot transfer since they only require a few examples (to select the correct path). Paths in $\Pi_4^{\text{NT},1}$ (Eq. 6) achieve non-perceptual transfer by reusing a module in the second layer, allowing applications to new input domains.

evaluating most of them involves training their new modules. This difficulty would be overcome if we could approximate a module composition's final performance without training its new modules.

Accordingly, our method divides the search space into subsets of module compositions which achieve different types of forward transfer and can be searched through separately. It then explores each subset using a subset-specific probabilistic model over the choice of pre-trained modules, designed to take advantage of the subset's properties. Querying each probabilistic model is efficient, as it does not involve training new parameters, which in turn enables a scalable search method.

Operationally, we first develop a probabilistic model over a set of module compositions which can achieve perceptual and few-shot transfer. The model exploits the fact that the input distribution on which a module is trained can indicate how successfully said module would process a set of inputs. Second, we identify a subset of module combinations capable of non-perceptual transfer and, using a new kernel, define a probabilistic model over this subset. We show that each of the two probabilistic models can be used to conduct separate searches through module combinations, which can then be combined into a scalable modular CL algorithm capable of perceptual, few-shot and non-perceptual transfer. Using two benchmark suites that evaluate different aspects of CL, we show that our approach achieves different types of knowledge transfer in large search spaces, is applicable to different input domains and modular neural architectures, and outperforms competitive baselines.

## 2 BACKGROUND

A continual learning algorithm is tasked with solving a sequence of problems $S = (\Psi_1, ..., \Psi_T)$, usually provided one at a time. We consider the supervised setting, in which each problem is characterised by a tuple $\Psi = (\mathcal{D}, \mathcal{T})$, where $\mathcal{D}$ is the input domain, comprised of an input space and an input distribution, and $\mathcal{T}$ is a task, defined by a label space and a labelling function (Pan & Yang, 2009). A CL algorithm aims to transfer knowledge between the problems in a sequence in order to improve each problem's generalisation performance. The knowledge transfer to a single problem can be defined as the difference in performance when the rest of the problems are not available.

CL algorithms have several desiderata. First, an algorithm should be *plastic*, i.e., learn to solve new problems. Second, it should be *stable* and avoid *catastrophic forgetting*. Third, it should be capable of *forward transfer*: the ability to transfer knowledge to a newly encountered problem. In particular, we distinguish between three types of knowledge transfer: between problems with similar input distributions (*perceptual*), between problems with different input distributions or different input-spaces (*non-perceptual*) and to problems with a few training examples (*few-shot*). Fourth, a CL algorithm should also be capable of *backward transfer*, meaning its performance on previously encountered problems should increase after solving new ones. Finally, the resource demands of a CL algorithm should *scale* sub-linearly with the number of solved problems.

Modular approaches represent a deep neural network $\zeta_\Theta$ as a composition of modules $\zeta_\Theta = m^L \circ \ldots \circ m^2 \circ m^1$, where the first module that is applied to the input is $m^1$. Each module $m^i$ represents a nonlinear transformation, parameterised by $\theta(m^i)$. It can consist of one or more hidden layers, each with a potentially different type (e.g. convolutional, fully connected, LSTM, Transformer, etc) and with a different activation function (e.g. ReLU or $tanh$). Given a problem, one needs to select optimal values for all parameters $\Theta$ and return the resulting neural network $\zeta_\Theta$ as the solution.

Modular approaches have been successfully applied to continual learning. After solving $t - 1$ problems, they accumulate a library of previously trained modules that each have been used to help perform one or more of the previous tasks. Denote this library by $\Lambda_t = \bigcup_{i=1}^{L} \{m_j^i\}_{j < t}$, where $m_j^i$ denotes a module in layer $i$ that was trained on the $j$-th problem. It is then possible to construct different modular neural networks by either selecting a pre-trained module from the library or by training a new one from scratch for each of the $L$ layers. We denote the set of all modular neural networks as $\Pi_t = \{(\{m_j^1\}_{j<t} \cup \{m_t^1\}) \times \ldots \times (\{m_j^L\}_{j<t} \cup \{m_t^L\})\}$, where $m_t^i$ denotes new module used in layer $i$ with randomly initialised parameters. We refer to each element of $\Pi_t$ as a path, as it can be seen as guiding the forward computation through different modules. Figure 1 illustrates the set of all paths $\Pi_4$ for the fourth problem of some sequence, using a modular architecture with 2 module layers.

Let a new problem $\Psi_t$ be specified by a training and a validation datasets, $\mathbf{D}^{\text{tr}} = (\mathbf{X}^{\text{tr}}, \mathbf{Y}^{\text{tr}})$ and $\mathbf{D}^{\text{val}} = (\mathbf{X}^{\text{val}}, \mathbf{Y}^{\text{val}})$. Evaluating each path $\pi$ requires training the network on $\mathbf{D}^{\text{tr}}$ and evaluating it on $\mathbf{D}^{\text{val}}$. Let $\pi^{[\text{pre}]}$ and $\pi^{[\text{new}]}$ denote the pre-trained and randomly initialised modules, respectively, from $\pi$. Training results in parameters $\Theta_{\text{new}}^* = \{\theta(m_j^i) : m_j^i \in \pi^{[\text{new}]}\}$ for each of the new modules. Parameters from the pre-trained modules are frozen. Thus, evaluating a path can be computationally expensive and one wants to evaluate as few paths as possible. However, the search space grows quickly. If there are $J$ pre-trained modules for each module types in the library, the search space consists of $(J + 1)^L$ unique paths. This necessitates a search strategy $\mathbb{S}(\Pi_t)$ which can prioritise the most promising paths for evaluation.

Modular algorithms can achieve many of the CL desiderata (Valkov et al., 2018). Catastrophic forgetting is prevented by freezing the parameters of all pre-trained modules in the library. A path with only randomly initialised parameters can be selected, ensuring the methods' plasticity. As more problems are solved, the pre-trained modules in the library can be reused in order to achieve all types of forward transfer, as discussed in the following sections. However, a key challenge has been to attain all transfer using a search strategy which scales to the enormous search space.

## 3 PROBABILISTIC MODULAR CONTINUAL LEARNING

We propose a probabilistic approach which uses the information available about the new problem to compute a distribution over different choices for pre-trained modules. The distribution does not model the new modules, and this allows efficient queries. This leads to a principled search strategy.

Specifically, we split $\Pi_t$ into subsets $\Pi_t^i \subset \Pi_t$ of paths in which the reused modules are at the same layer positions. For each subset, we define a a probability distribution over the choice of pre-trained modules, $p(\pi^{[\text{pre}]}|\mathbf{X}^{\text{tr}}, \mathbf{Y}^{\text{tr}}, \mathbf{X}^{\text{tr}}, \mathbf{Y}^{\text{val}}, E_j)$. This distribution uses the available training and validation datasets, and the previously evaluated paths $E_j$. Here, $j$ denotes the index of the current path suggestion. This distribution can be queried efficiently, as it does not involve randomly initialised modules. Now we define a search strategy for each path subset $\Pi_t^i$. At step $j$, this strategy selects the unevaluated path with the highest posterior probability of its pre-trained modules:

$$\mathbb{S}_{\text{MAP}}(\Pi_t^i) = \left( \underset{\pi \in \Pi_t^i}{\arg\max}\, p(\pi^{[\text{pre}]}|\mathbf{X}^{\text{tr}}, \mathbf{Y}^{\text{tr}}, \mathbf{X}^{\text{tr}}, \mathbf{Y}^{\text{val}}, E_j) \right)_j \tag{1}$$

Multiple subsets of $\Pi_t$ can be explored by applying this search strategy to each sequentially.

Next, we first apply our approach to subsets of paths which are capable of perceptual and few-shot transfer. Second, we apply it to a subset of paths capable of non-perceptual transfer. In both cases, we define a suitable Bayesian model, which takes advantage of each subset's properties. Finally, we show how the two search strategies can be combined.

## 4 EFFICIENT PERCEPTUAL TRANSFER AND FEW-SHOT TRANSFER

To achieve perceptual transfer, a model needs to transfer knowledge on how to transform the input. For a path, this means reusing the first $l \in \{1, ..., L\}$ module layers. Additionally, a path that reuses all $L$ layers can allow for few-shot transfer, since there are no new parameters that need to be learned. Therefore, we start by considering paths where the first $l$ layers are selected from the library, while the rest are selected to be new randomly initialised modules, i.e.

$$\Pi_t^{\text{PT},l} = \{\pi : \pi = \{m_{<t}^i, i \leq l\} \cup \{m_t^i, i > l\}\}. \tag{2}$$

Each subset grows polynomially with the size of the library and exponentially with the number of layers, which makes a naive search inapplicable. Several previous papers consider this search space. For example, MNTDP (Veniat et al., 2020) only consider paths where all of the pre-trained modules come from the same previous solution. This leads to a rigid approach which cannot compose pre-trained modules in novel ways. In contrast, LMC (Ostapenko et al., 2021) propose to model each module's input density, enabling flexible input-specific composition of modules. However, their approximations double the storage requirements and requires that all modules be loaded in memory, which could hinder scaling to a large number of pre-trained modules.

In this section, we present a probabilistic model, which allows us to efficiently explore novel input-specific module compositions. In contrast to LMC, we efficiently approximate each module's input density and also allow for prior knowledge over the choice of pre-trained modules to be incorporated. Evaluating our approximations can be done without loading the corresponding pre-trained modules in memory and we show that this leads to a scalable search strategy.



Figure 2: A graphical model of the joint distribution over a PT path with three pre-trained modules $\pi = (M^1, M^2, M^3)$ and hidden states $H_i^1$ and $H_i^2$.

Our motivating intuition is to select pre-trained modules so that the distribution of inputs each module receives in the new problem will match the input distribution said module was trained on. This minimizes the amount of distribution shift faced by each module, hopefully increasing the performance of the full network. Using this insight, for a given value of $l$, we define a probabilistic model over the choices of pre-trained modules and their $N$ inputs (including $X$ - the problem inputs, and $H^i$ - relevant hidden states). Figure 2 shows the corresponding graphical model for $l = 3$. The hidden states and the choice of pre-trained modules are latent variables which are sequentially used to generate the observed inputs. This allows us to infer the distribution over pre-trained modules, given the observed inputs:

$$p(M^1, ..., M^l, X_1, ..., X_N, H_1^1, ..., H_N^1, ..., H_1^{l-1}, ..., H_N^{l-1})$$
$$= \prod_{j=1}^{N} \left\{ p(X_j | H_j^1, M^1) \prod_{i=2}^{l-1} \left[ p(H_j^{i-1} | H_j^i, M^i) \right] p(H_j^{l-1} | M^l) \right\} \prod_{i=1}^{l} \left[ p(M^i) \right]. \tag{3}$$

As a result, we then model the posterior of a PT path as only being dependent on the training inputs, $p(\pi^{[\text{pre}]} | \mathbf{X}^{\text{tr}}) := p(m^1, ..., m^l | \mathbf{x}_1, ..., \mathbf{x}_N)$, and express it in terms of quantities which we can approximate, as derived in Appendix A:

$$p(m^1, ..., m^l | \mathbf{x}_1, ..., \mathbf{x}_N) \propto p(m^1, ..., m^l, \mathbf{x}_1, ..., \mathbf{x}_N) = p(m^1, ..., m^l, \mathbf{h}_1^0, ..., \mathbf{h}_N^0)$$
$$\approx \prod_{j=1}^{N} \left\{ \prod_{i=1}^{l-1} \left[ \frac{q(\mathbf{h}_j^{i-1} | m^i)}{\sum_{m^{i+1'}} q(\mathbf{h}_j^i | m^{i+1'}) p(m^{i+1'})} \right] q(\mathbf{h}_j^{l-1} | m^l) \right\} \prod_{i=1}^{l} p(m^i) \tag{4}$$

where $q(\mathbf{h}_j^{i-1} | m^i)$ denotes our approximation to the training input distribution of the module $m^i$.

We define a prior distribution over pre-trained modules which gives preference to modules which helped achieve a higher accuracy, on the problems which they have been trained on. Our motivation is that such modules have learned a more generalisable transformation of their inputs. We use softmax to define a prior, $p(M^i)$, over the modules in layer $i$ which is proportional to each module's previously achieved accuracy. The prior is defined in Appendix B, eq. 12.

We set out to construct an approximation of a module's training input distribution, $p(H^{i-1}|M^i)$, which is easy to compute in terms of implementation, storage and computational resources. After a module is trained, we perform dimensionality reduction of each sample from the module's training distribution. We use a random projection (Johnson, 1984) to reduce each sample's dimensionality from $c$ to $k \ll c$. Finally, we compute the sample mean and covariance of the projected data samples in order to approximate their distribution using a multivariate Gaussian. This greatly reduces the number of parameters required to approximate the resulting distribution from $c + c^2$ to $k + k^2$. When maximising the posterior using Eq. 4, we compute $q(\mathbf{h}_j^{i-1}|m^i) \propto q(\mathbf{A}^{i-1}\mathbf{h}_j^{i-1}|m^i)$, where $\mathbf{A}^{i-1}$ is a layer-specific randomly generated matrix used for random projection (Pedregosa et al., 2011). Surprisingly, our ablation experiments, presented in Appendix E.2, suggest that using a Gaussian approximation over randomly projected inputs is also more reliable than using a Gaussian approximation over the original inputs.

**Search Strategy**   The number of possible PT paths of length $l$ increases exponentially with $l$. As a result, evaluating $p(\pi|\mathbf{X}^{\text{tr}})$ for all PT paths in the $\mathbb{S}_{\text{MAP}}$ search strategy (Eq. 1) is infeasible. Instead, we augment the search with a greedy policy: after selecting the first $a$ pre-trained modules, we freeze this selection and reuse the same $a$ modules for PT paths with $l > a$ which reuse more pre-trained modules. Also, for each value of $l$, the search only recommends the path which has the highest probability under our model, since this is the path which is best equipped to handle the inputs. We define the resulting search strategy over all $\Pi^{\text{PT}}$ as:

$$\mathbb{S}_{\text{G}}^{\text{PT}}(\Pi_t^{\text{PT}}) := \{\pi^{*\text{PT},l}\}_{l=1}^{L} \tag{5}$$
$$\text{where } \pi^{*\text{PT},l} = \pi^{*\text{PT},l-1}[: l-1] \cup m^{*,l} \cup \{m_t^i\}_{i=l}^{L}$$
$$m^{*,l} = \arg\max_{m^l} p(\pi^{*'\text{PT},l-1}[: l-1] \cup m^l|\mathbf{x}_1, ..., \mathbf{x}_N).$$

It states that in order to select the most promising PT path with $l$ pre-trained modules, $\pi^{*\text{PT},l}$, we choose the previously selected $l-1$ pre-trained modules, $\pi^{*\text{PT},l-1}[: l-1]$, and append the optimal choice for layer $l$, $m^{*,l}$, which maximises the resulting posterior distribution.

The storage requirement of this search is associated with the number of modules in the library. Therefore, it scales linearly with the number of solved problems, $\mathcal{O}(t)$. The exact number depends on the type of similarity between the encountered problems. If perceptual transfer and few-shot transfer is often possible, the number of modules in the library will grow more slowly. Our greedy search strategy requires that each module from the library is loaded at most once, which leads to a constant memory requirement. The computational complexity of this search remains constant with the number of solved problems since it selects a constant number of paths for each problem: $L + 1$ ($L$ suggested by our search strategy, plus a randomly initialised model).

## 5   EFFICIENT NON-PERCEPTUAL TRANSFER

To achieve non-perceptual transfer, an algorithm can use pre-trained modules for the last $l$ module layers. This represents knowledge on how to transfer a latent representation of the input to a task-specific prediction. Therefore, we identify subsets $\Pi_t^{NT,l} \in \Pi_t$ which consists of paths capable of non-perceptual transfer. Each path has its first $L - l$ modules randomly initialised while the rest $l$ modules are selected from a library:

$$\Pi_t^{\text{NT},l} = \left\{\pi_t^{\text{NT},l} : \pi_t^{\text{NT},l} = \{m_t^i\}_{i=1}^{l} \cup \{m_{<t}^i\}_{i=L-l+1}^{L}\right\}. \tag{6}$$

**Probabilistic Model**   While the pre-trained modules in PT paths process the inputs, the pre-trained modules in NT paths predict the outputs. As a result, we build a probabilistic model which captures how good an NT path is in predicting the correct outputs after training. Concretely, for NT paths with the same number $l$ of pre-trained modules, we model the posterior over the choice of pre-trained modules given the validation dataset as:

$$p(\pi^{[\text{pre}]}|\mathbf{X}^{\text{val}}, \mathbf{Y}^{\text{val}}) \propto p(\mathbf{Y}^{\text{val}}|\mathbf{X}^{\text{val}}, \pi^{[\text{pre}]})p(\pi^{[\text{pre}]}). \tag{7}$$

We define a prior over $\pi^{[\text{pre}]}$ which assigns equal non-zero values only to pre-trained modules which have been used together to solve a previous problem. This reflects our prior assumption that using novel combination of modules for non-perceptual transfer is unnecessary for our sequences.

We model $p(\mathbf{Y}^{\text{val}}|\mathbf{X}^{\text{val}}, \pi^{[\text{pre}]})$ using a Gaussian process (GP), which before recommending a path at step $j$, is fit on previous evaluations $E_j$. For numerical stability, we model the logarithm, predicting:

$$\log p(\mathbf{Y}^{\text{val}}|\mathbf{X}^{\text{val}}, \pi_i^{[\text{pre}]}) \sim GP(\mathbf{0}, \kappa(\pi_i^{[\text{pre}]}, \pi_j^{[\text{pre}]})) \tag{8}$$

To enable this, we define a kernel function $\kappa$ between the pre-trained modules of two NT paths, based on the standard Euclidean distance in function space. Let the inner product between $f : \Omega \to \mathbb{R}^r$ and $g : \Omega \to \mathbb{R}^r$ be: $\langle f \ , \ g \rangle = \int_\Omega f(\mathbf{z}) \cdot g(\mathbf{z})d\mathbf{z}$ . This allows us to compute the distance between two functions as:

$$d(f,g) := ||f - g|| = \sqrt{\langle f - g \ , \ f - g \rangle} = \sqrt{\int_\Omega (f(\mathbf{z}) - g(\mathbf{z})) \cdot (f(\mathbf{z}) - g(\mathbf{z}))d\mathbf{z}} \ .$$

We approximate this value using Monte Carlo integration with a set of inputs, $Z$, from the functions' common input space. We can then define the kernel function between the chosen pre-trained modules of two NT paths with the same $l$ value, using the squared exponential kernel function and the distance between the functions computed by their last $l$ modules:

$$\kappa(\pi_i^{[\text{pre}]}, \pi_j^{[\text{pre}]}; Z) = \sigma^2 \exp\left\{-(d(\pi_i^{[\text{pre}]}, \pi_j^{[\text{pre}]}; Z)^2)/(2\gamma^2)\right\} \tag{9}$$

where $\sigma$ and $\gamma$ are the kernel hyperparameters which are fit to maximise the marginal likelihood of a GP's training data (Rasmussen & Williams, 2006).

To create $Z$ for a value of $l$, we store a few samples from the input distribution of each pre-trained module at layer $L - l + 1$. Given a new problem, we create $Z$ by combining all the stored hidden activations. This allows us to compare functions on regions of the input space with higher density.

**Search Strategy**    We modify $\mathbb{S}_{\text{MAP}}$ and apply it on a single subset $\Pi_t^{\text{NT},l_{\min}}$ where $l_{\min}$ is the user-defined minimum number of modules which need to be transferred. Due to our choice of prior, maximising the posterior using Eq. 7 amounts to maximising the log-likelihood over paths with nonzero probability. However, our GP-based approximation provides a distribution of the likelihood's value (reflecting the uncertainty of the GP's prediction). To account for this, we use an acquisition function, namely Upper Confidence Bound (UCB) (Shahriari et al., 2015) which provides an optimistic prediction of the likelihood. We compute it as $UCB(\mu, \sigma) = \mu + \beta\sigma$, where $\beta$ is a hyperparameter, and $\mu$ and $\sigma$ are respectively the mean and the standard deviation of the GP-predicted distribution. The resulting search strategy resembles Bayesian Optimisation:

$$\mathbb{S}_{\text{BO-MAP}}(\Pi_t^{\text{NT},l_{\min}}) = \left(\underset{\pi \in \Pi_t^i}{\arg\max} \ UCB\left(\log p(\mathbf{Y}^{\text{val}}|\mathbf{X}^{\text{val}}, \pi^{[\text{pre}]})\right)\right)_j . \tag{10}$$

At each step $j$, we fit a GP (Eq. 8) with a kernel $\kappa$ (Eq. 9) on the set of evaluations of the previous paths $E_j$, and select the unevaluated path $\pi$ whose pre-trained modules maximise the UCB.

The GP makes it possible to detect when further improvement is unlikely, allowing us to perform early stopping. For this purpose we compute the Expected Improvement, $EI\left(\log p(\pi^{[\text{pre}]}|\mathbf{X}^{\text{val}}, \mathbf{Y}^{\text{val}})\right)$ of the path selected for evaluation at each step $j$. If it is lower than a certain threshold, we exit the search and do not recommend any more paths for evaluation. EI-based early stopping has been previously suggested in Nguyen et al. (2017) and, similarly to Makarova et al. (2022), our preliminary experiments showed that it leads to fewer evaluations, compared to using UCB for early stopping.

In the worst case $\mathbb{S}_{\text{BO-MAP}}$ would propose $\mathcal{O}(t - 1)$ paths for evaluation which scales linearly with the number of solved problems. While early stopping reduces the number of evaluated paths, our experiments suggest said number still scales linearly with the number of problems. However, our ablation experiments, presented in Appendix E, demonstrate that our search strategy exhibits a good anytime performance. This means that our Bayesian optimisation algorithm is capable of quickly finding a path which achieves positive non-perceptual transfer. Therefore, it is possible to constrain $\mathbb{S}_{\text{BO-MAP}}$ to evaluating a constant number of paths and still achieve non-perceptual transfer across a long sequence of problems.

| | | SA | MCL-RS | HOUDINI | MNTDP-D | PeCL, CCL |
|---|---|---|---|---|---|---|
| $\hat{\mathcal{A}}$ | $S^{\text{few}}$ | 75.47 | 78.14 | 80.82 | 82.18 | **88.12** |
| | $S^{\text{out}}$ | 74.25 | 76.16 | 74.40 | 77.95 | **78.15** |
| | $S^{\text{out*}}$ | 72.27 | 73.39 | 72.27 | 75.48 | **75.72** |
| | $S^{\text{out**}}$ | 71.51 | 73.85 | 71.75 | 73.71 | **75.73** |
| | $S^{\text{pl}}$ | 93.61 | 93.63 | 93.61 | 93.72 | **93.79** |
| | $S^-$ | 73.88 | 76.67 | 79.59 | 81.67 | **81.92** |
| | $S^+$ | 73.61 | **75.08** | 73.61 | 74.54 | 74.49 |
| | Aggregated: | 76.37 | 78.13 | 78.01 | 79.89 | **81.13** |
| $Tr^{-1}$ | $S^{\text{few}}$ | 0. | 5.87 | 4.54 | 11.42 | **46.07** |
| | $S^-$ | 0. | 17.22 | 34.27 | **34.29** | **34.29** |
| | $S^{\text{out}}$ | 0. | 5.64 | 0. | **15.41** | **15.41** |
| | $S^{\text{out*}}$ | 0. | 0.43 | 0. | **12.53** | **12.53** |
| | $S^{\text{out**}}$ | 0. | 4.61 | 1.46 | 1.74 | **12.04** |
| | $S^{\text{pl}}$ | 0. | 0. | 0. | **00.20** | **00.20** |
| | $S^+$ | 0. | 0. | 0. | 0. | 0. |
| | Aggregated: | 0. | 4.82 | 5.75 | 10.8 | **17.22** |

Table 1: Results on compositional benchmarks that do not evaluate non-perceptual transfer.

| | | SA | MCL-RS | HOUDINI | MNTDP-D | PeCL | NoCL | CCL |
|---|---|---|---|---|---|---|---|---|
| $\hat{\mathcal{A}}$ | $S^{\text{in}}$ | 89.01 | 90.85 | 89.32 | 90.62 | 90.26 | 92.20 | **92.82** |
| | $S^{\text{sp}}$ | 87.94 | 92.22 | **92.99** | 87.94 | 87.92 | 91.92 | 91.93 |
| | Aggregated: | 88.48 | 91.54 | 91.16 | 89.28 | 89.09 | 92.06 | **92.38** |
| $Tr^{-1}$ | $S^{\text{in}}$ | 0. | 1.81 | 11.04 | 9.70 | 7.61 | 18.89 | **22.28** |
| | $S^{\text{sp}}$ | 0. | 25.68 | **30.27** | 0. | 0. | 23.65 | 23.65 |
| | Aggregated: | 0. | 13.75 | 20.66 | 4.85 | 3.805 | 21.27 | **22.97** |

Table 2: Results on compositional benchmarks that evaluate non-perceptual transfer.

# 6 EXPERIMENTS

We evaluate the perceptual, non-perceptual and few-shot transfer capabilities, as well as the scalability, of our methods. Also, we assess the applicability of our methods to disparate input domains and neural architectures. To this end, we perform experiments on a new benchmark suite as well as on the CTrL (Veniat et al., 2020) benchmark suite.

We derive three modular CL algorithms: PeCL, which can perform perceptual and few-shot transfer using $\mathbb{S}_G^{\text{PT}}(\Pi_t^{\text{PT}})$ (Eq 5); NoCL, which can perform non-perceptual transfer using $\mathbb{S}_{\text{BO-MAP}}(\Pi_t^{\text{NT},l_{\min}})$ (Eq. 10); and CCL, which combines both search strategies using $\mathbb{S}_{\text{CCL}}(\Pi_t^{\text{PT}} \cup \Pi_t^{\text{NT},l_{\min}}) :=$ $S_G^{\text{PT}}(\Pi_t^{\text{PT}}) + S_{\text{BO-MAP}}(\Pi_t^{\text{NT},l_{\min}})$. We compare our algorithms to a number of competitive modular CL baselines: Standalone (SA), which trains a new model for every problem; MCL-RS, which randomly selects paths from $\Pi_t$ (shown to be a competitive baseline in high-dimensional search spaces and in neural architecture search (Li & Talwalkar, 2020)); HOUDINI(Valkov et al., 2018), with a fixed neural architecture to keep the results comparable; and MNTDP-D (Veniat et al., 2020).

We use the same set of hyperparameters, listed in Appendix D, for our approaches across both benchmark suites, which suggests that this choice of hyperparameters is robust and applicable to different problems and neural architectures. Moreover, we use a problem-specific random seed to make the training process deterministic, so that the difference in performance can be accredited only to the LML algorithm, and not to randomness introduced during training. For each baseline, we assess the performance on a held-out test dataset and report the average accuracy of the final model across all problems, $\hat{\mathcal{A}}$ (see Eq. 14), as well as the amount of forward transfer on the last problem, $Tr^{-1}$, computed as the difference in accuracy, compared to the standalone baseline (see Eq. 15).

**Compositional Benchmarks.** We introduce a benchmark suite consisting of different sequences which can diagnose different CL properties. We build on the CTrL benchmarks (Veniat et al., 2020) by instead using problems with compositional tasks, in which each problem combines an image clas-

sification task with a two-dimensional pattern recognition task. This allows us to design sequences to evaluate specific desiderata, such as non-perceptual transfer, few-shot transfer, and scalability to long problem sequences. We have the following sequences of length 6. In $S^{\text{pl}}$, each problem is represented by a large dataset in order to evaluate an algorithm's plasticity. In $S^+$ the last problem is the same as the first, but is represented with a larger dataset, allowing for backward transfer to be evaluated. Conversely, in $S^-$, the first and last problems are also the same but the first is represented by a bigger dataset, thus, evaluating forward knowledge transfer to the same problem. We define three different sequences which evaluate perceptual transfer. In $S^{\text{out}}$, the input domains of the first and last problems are the same. In $S^{\text{out}*}$ the input domains of the first, second and last problems are the same, making it harder to decide which problem to transfer from. In $S^{\text{out}**}$ the first and last problems are the same, however, perceptual knowledge needs to be transferred from the second problem, as it also has the same domain but is represented by a larger dataset. $S^{\text{few}}$ evaluates few-shot transfer, since the last problem is represented by 10 data points and solving it requires re-combining knowledge from problems 2 and 4 in a novel way. Finally, there are two sequences evaluating non-perceptual transfer. In $S^{\text{in}}$ the first and last problem share the same two-dimensional pattern, while having different input image domains. $S^{\text{sp}}$ is the same as $S^{\text{in}}$, except last problem's input space is also re-shaped from an image to a one-dimensional vector, which loses the structural information and requires different architecture for the first few modules in order to process the input. Therefore, this evaluate an algorithm's ability to transfer across different input spaces and similar but not identical neural architectures. Finally, $S^{\text{long}}$ is a sequence of 60 randomly selected problems, which evaluates perceptual and non-perceptual transfer on long sequences.

The neural architecture which we use consists of $L = 8$ modules, which leads to a large search space, since even for $t = 6$, there are $\mathcal{O}(t^L = 1679616)$ different possible paths. Full specification of the problems, sequences, neural architecture and training procedure can be found in Appendix D. We create 3 realisations of each sequence by randomly selecting different compositional problems. Afterwards, for each sequence, we report the measurements, averaged over these 3 versions.

**Results.** Overall, CCL has higher performance than the other methods, averaged across sequences (Tables 1 and 2). Our method is designed specifically to enhance perceptual, few-shot and non-perceptual transfer. Therefore, we expect its performance on other CL desiderata to be comparable to the baselines. We find that PeCL and CCL exhibit the same performance for sequences which do not involve non-perceptual transfer (Table 1). Our method demonstrates similar plasticity to the others, according to $S^{\text{pl}}$, as well as similar performance on $S^+$ which evaluates backward transfer. Note that $\mathbb{S}_G^{\text{PT}}$ enables our algorithms to outperform the baselines on the sequences which require forward knowledge transfer, with the other scalable modular algorithm, MNTDP-D, being second. On $S^{\text{out}**}$, our performance-based prior helps our method identify the correct modules to be transferred, leading to a $+10.3$ higher accuracy on the last problem than MNTDP-D. On $S^{\text{few}}$ our algorithm's ability to perform few-shot transfer leads to $+34.7$ higher accuracy on the last problem than MNTDP-D.

For non-perceptual transfer (Table 2), we find that $\mathbb{S}_{\text{BO-MAP}}$ enables NoCL and CCL to transfer knowledge across different input distributions and input spaces. In $S^{sp}$, the different input space necessitates a different modular architecture for the first 5 modules, resulting in a much smaller search space, $\mathcal{O}(6^3 = 216)$. This allows non-scalable approaches, namely MCL-RS and HOUDINI, to also be effective on this sequence. Lastly, CCL's performance demonstrates that combining $\mathbb{S}_G^{\text{PT}}$ and $\mathbb{S}_{\text{BO-MAP}}$ leads to a better performance on sequences that allow for both perceptual and non-perceptual transfer. Finally we evaluated our approach on $S^{\text{long}}$, a sequence of 60 problems. PeCL achieved $+7.37$ higher average accuracy than the standalone baseline demonstrating its ability to achieve perceptual transfer on a long sequence of problems. MNTDP-D achieved $8.83$ higher average accuracy than SA which confirmed its scalability. Finally, CCL performed the best, attaining $+12.25$ higher average accuracy than SA. This shows that our approach can successfully attain perceptual and non-perceptual transfer across a long sequence of problems.

**CTrL benchmarks.** The CTrL benchmark suite (Veniat et al., 2020) defines fewer sequences to evaluate different CL properties. Namely, they specify $S^{\text{pl}}$, $S^+$, $S^-$, $S^{\text{out}}$, $S^{\text{sp}}$ which are defined identically to ours. In contrast, the sequences are over multi-class classification tasks of coloured images from different domains. They also use a different modular architecture, based on ResNet18 (He et al., 2016), which is more complex than the architecture used in the previous benchmarks. Our experimental setup, detailed in Appendix D, mirrors the one used in Veniat et al. (2020), except that we make the training process deterministic, as discussed above.

| | | SA | MNTDP-D | PeCL | NoCL | CCL |
|---|---|---|---|---|---|---|
| $\hat{\mathcal{A}}$ | $S^{\text{in}}$ | 58.77 | 61.36 | 61.78 | **63.41** | 63.10 |
| | $S^{\text{out}}$ | 74.25 | 77.95 | **78.15** | - | **78.15** |
| | $S^{\text{pl}}$ | 58.25 | 93.72 | **93.79** | - | **93.79** |
| | $S^-$ | 56.28 | 81.67 | **81.92** | - | **81.92** |
| | $S^+$ | 73.61 | **74.54** | 74.49 | - | 74.49 |
| | Aggregated: | 64.23 | 77.85 | 78.03 | - | **78.29** |
| $Tr^{-1}$ | $S^{\text{in}}$ | 0. | 22.12 | 24.67 | **32.57** | **32.57** |
| | $S^{\text{out}}$ | 0. | **15.41** | **15.41** | - | **15.41** |
| | $S^{\text{pl}}$ | 0. | **00.20** | **00.20** | - | **00.20** |
| | $S^-$ | 0. | **34.29** | **34.29** | - | **34.29** |
| | $S^+$ | 0. | 0. | 0. | - | 0. |
| | Aggregated: | 0. | 14.40 | 14.91 | - | **16.49** |

Table 3: The evaluations on the CTrL sequences, except for $S^{\text{long}}$. For each sequence, we report average accuracy $\mathcal{A}$ and the amount of forward transfer on the last problem $Tr^{-1}$.

Our results (Table 3) demonstrate that our methods, namely PeCL and CCL, achieve similar performance to MNTDP-D on $S^{\text{pl}}$, $S^+$, $S^-$, $S^{\text{out}}$ which evaluate plasticity, backward and perceptual transfer. On $S^-$, NoCL and CCL successfully perform non-perceptual transfer, leading to superior performance on the last problem of the sequence ($+10.45$ higher than MNTDP). CTrL also specifies $S^{\text{long}}$ which has 100 problems but only evaluates perceptual transfer. PeCL was successfull in transferring knowledge, achieving an average accuracy that was $+14.48$ higher than SA. Overall, the results demonstrates that both $\mathbb{S}_{\text{BO-MAP}}$ are also applicable to more complex modular architectures.

## 7 RELATED WORK

This work considers the data-incremental (De Lange et al., 2021) supervised setting of continual learning. Other settings can involve overlapping problems Farquhar & Gal (2018) or reinforcement learning (Khetarpal et al., 2020). CL desidarata is also presented in Schwarz et al. (2018); Hadsell et al. (2020); Delange et al. (2021), which include other properties, e.g. selective forgetting of trivial information, but do not distinguish between different types of forward transfer. We list such types following Valkov et al. (2018), but add the requirement of non-perceptual transfer over input spaces.

Continual learning methods can be categorised into ones based on regularisation, replay or a dynamic architecture (Parisi et al., 2019). The first two share the same set of parameters across all problems which limits their capacity and, in turn, their plasticity (Kirkpatrick et al., 2017). Dynamic architecture methods can share different parameters by learning a problem-specific parameter masks (Mallya & Lazebnik, 2018) or adding more parameters (Rusu et al., 2016). This category includes modular approaches to CL, which share and introduce new modules, allowing groups of parameters to be trained and always reused together. Modular approaches mainly differ by their search space and their search strategy. PathNet (Fernando et al., 2017) uses evolutionary search to search through paths that combine up to $4$ modules per layer. Rajasegaran et al. (2019) use random search on the set of all paths. HOUDINI (Valkov et al., 2018) uses type-guided exhaustive search on the set of all possible modular architectures and all paths. This method can attain the three types of forward transfer, but does not scale to large search spaces. Their results also show that exhaustive search leads to better performance than evolutionary search. MNTDP-D (Veniat et al., 2020) is a scalable approach which restrict its search space to perceptual transfer paths, derived from previous solutions. Similarly to $\mathbb{S}_G^{\text{PT}}$, MNTDP-D evaluates only $L + 1$ paths per problem, however, the approach does not allow for novel combinations of pre-trained modules which prevents it from achieving few-shot transfer. In contrast, our approach can achieve all three types of forward transfer. LMC (Ostapenko et al., 2021) makes a soft selection over paths. For each layer, they compute a linear combination of the outputs of all available pre-trained modules. To do this, the authors model the distribution over each module's outputs using a separate auto-encoder. In contrast, for $\mathbb{S}_G^{\text{PT}}$ we use multivariate Gaussian approximations of the projected inputs of a pre-trained module, which requires orders of magnitude fewer parameters. LMC can achieve perceptual and few-shot transfer, but is not capable of non-perceptual transfer. Finally, LMC requires that all modules be kept in memory which limits it applicability to larger libraries, thus, larger search spaces.

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

## A DERIVING THE POSTERIOR FOR PT PATHS

In this section we show how we approximate the posterior distribution $p(m^1_{<t}, ..., m^l_{<t} | \mathbf{x}_1, ..., \mathbf{x}_N)$. To ease the presentation, and without loss of generality, we set the number of pre-trained modules $l = 3$. Next, we express the joint distribution in terms of quantities which we can approximate.

$$p(M^1, M^2, M^3, X_1, ..., X_N, H^1_1, ..., H^1_N, H^2_1, ..., H^2_N)$$
$$= p(M^1)p(M^2)p(M^3) \prod_{j=1}^{N} p(X_j | H^1_j, M^1) p(H^1_j | H^2_j, M^2) p(H^2_j | M^3)$$

Here, $p(X_j | H^1_j, M^1)$ can be expressed as:

$$p(X_j | H^1_j, M^1) = \frac{p(X_j, H^1_j, M^1)}{p(H^1_j, M^1)} = \frac{p(H^1_j | X_j, M^1)p(X_j | M^1)p(M^1)}{p(H^1_j)p(M^1)} = \frac{p(H^1_j | X_j, M^1)p(X_j | M^1)}{\sum_{m^{2'}} p(H^1_j | m^{2'})p(m^{2'})}.$$

Moreover, $p(H^1_j | H^2_j, M^2)$ can be expressed as:

$$p(H^1_j | H^2_j, M^2) = \frac{p(H^1_j, H^2_j, M^2)}{p(H^2_j, M^2)} = \frac{p(H^2_j | H^1_j, M^2)p(H^1_j | M^2)p(M^2)}{p(H^2_j)p(M^2)}$$
$$= \frac{p(H^2_j | H^1_j, M^2)p(H^1_j | M^2)}{\sum_{m^{3'}} p(H^2_j | m^{3'})p(m^{3'})}.$$

Therefore, the joint distribution can be expressed as:

$$p(M^1, M^2, M^3, X_1, ..., X_N, H^1_1, ..., H^1_N, H^2_1, ..., H^2_N)$$
$$= p(M^1)p(M^2)p(M^3) \prod_{j=1}^{N} \frac{p(H^1_j | X_j, M^1)p(X_j | M^1)}{\sum_{m^{2'}} p(H^1_j | m^{2'})p(m^{2'})} \frac{p(H^2_j | H^1_j, M^2)p(H^1_j | M^2)}{\sum_{m^{3'}} p(H^2_j | m^{3'})p(m^{3'})} p(H^2_j | M^3).$$

$$(11)$$

This expression contains three groups of distributions, which we need to define. First, we can define a prior over the choices of pre-trained modules for different module layers, $p(M^i)$. Second, we can approximate each module's training input distribution, leading to $p(H^i_j | m^{i+1}_{<t}) \approx q(H^i_j | m^{i+1}_{<t})$. The third group of distributions contains $p(H^i_j | H^{i-1}_j, M^i)$ which is a distribution over the values of a hidden layer $H^i$ given the previous hidden layer $\mathbf{h}^{i-1}$ and a module $m^i$. However, said hidden layer is given by the deterministic transformation $\mathbf{h}^i_j = m^i(\mathbf{h}^{i-1}_j)$. Therefore, we can model $p(H^i_j | H^{i-1}_j, M^i) = \delta(H^i_j - \mathbf{h}^i_j)$ using the Dirac delta function $\delta$. This function has the property that $\int_{-\infty}^{\infty} f(z)\delta(z - c)dz = f(c)$, which we use next in order to simplify the posterior. We write:

$$p(m^1, m^2, m^3 | \mathbf{x}_1, ..., \mathbf{x}_N) = \frac{p(m^1, m^2, m^3, \mathbf{x}_1, ..., \mathbf{x}_N)}{p(\mathbf{x}_1, ..., \mathbf{x}_N)} \propto p(m^1, m^2, m^3, \mathbf{x}_1, ..., \mathbf{x}_N)$$

$$= \int_{-\infty}^{\infty} ... \int_{-\infty}^{\infty} p(m^1, m^2, m^3, \mathbf{x}_1, ..., \mathbf{x}_N, \mathbf{h}'^1_1, ..., \mathbf{h}'^1_N, \mathbf{h}'^2_1, ..., \mathbf{h}'^2_N)d\mathbf{x}_1...d\mathbf{h}'_{\mathbf{N}}$$

$$= p(m^1)p(m^2)p(m^3) \prod_{j=1}^{N} \frac{p(x_j | m^1)}{\sum_{m^{2'}} p(\mathbf{h}^1_j | m^{2'})p(m^{2'})} \frac{p(\mathbf{h}^1_j | m^2)}{\sum_{m^{3'}} p(\mathbf{h}^2_j | m^{3'})p(m^{3'})} p(\mathbf{h}^2_j | m^3)$$

$$\approx p(m^1)p(m^2)p(m^3) \prod_{j=1}^{N} \frac{q(\mathbf{x}_j | m^1)}{\sum_{m^{2'}} q(\mathbf{h}^1_j | m^{2'})p(m^{2'})} \frac{q(\mathbf{h}^1_j | m^2)}{\sum_{m^{3'}} q(\mathbf{h}^2_j | m^{3'})p(m^{3'})} q(\mathbf{h}^2_j | m^3)$$

In general, for $l$ pre-trained modules, we approximate the numerator of the posterior using:

$$p(m^1, ..., m^l | \mathbf{x}_1, ..., \mathbf{x}_N) \propto p(m^1, ..., m^l, \mathbf{x}_1, ..., \mathbf{x}_N)$$

$$\approx \prod_{i=1}^{l} p(m^i) \prod_{j=1}^{N} \left\{ \prod_{i=1}^{l-1} \left[ \frac{q(\mathbf{h}_j^{i-1} | m^i)}{\sum_{m^{i+1'}} q(\mathbf{h}_j^i | m^{i+1'}) p(m^{i+1'})} \right] q(\mathbf{h}_j^{l-1} | m^l) \right\}$$

To compute this, we need to define a prior distribution over modules and approximate a module's training input distribution.

## B  DEFINING THE PRIOR FOR PT PATHS

Computing Eq. 4 requires us to define a prior distribution over the choice of a pre-trained module, $p(M^i)$. Say two modules $m_a^i$ and $m_b^i$ are trained using two different paths on two different problems. Now, say that the model trained on problem $a$ achieved $Acc(a)$ validation accuracy, while the model trained on problem $b$ achieved a higher validation accuracy $Acc(b) = Acc(a) + \delta$, for $\delta > 0$. We hypothesise that the module, whose model achieved the higher accuracy after training, is likely to compute a more useful transformation of its input. Therefore, if $m_a^i$ and $m_b^i$ have a similar likelihood for a given set of training data points, we would give preference to using $m_b^i$. To this end, we define the prior distribution in terms of a module's original accuracy using the *softmax* function as follows:

$$p(m_j^i) = \frac{e^{\frac{Acc(j)}{T}}}{\sum_{m_{j'}^i} e^{\frac{Acc(j')}{T}}}. \tag{12}$$

Here $T$ is the temperature hyperparameter which we compute as follows. Suppose that, for a given set of inputs $\{\mathbf{x}_1, ..., \mathbf{x}_N\}$, we have selected the first $i-1$ modules and have computed the inputs to the $i$th module, $D^{H^{i-1}} = \{h_j^{i-1}\}_{j=1}^N$. Moreover, suppose that the likelihood of module $m_a^i$ is slightly higher than the likelihood of $m_b^i$, i.e. that $p(D^{H^{i-1}} | m_a^i) > p(D^{H^{i-1}} | m_b^i)$. However, because the model of $m_b^i$ was trained to a higher accuracy ($Acc(b) = Acc(a) + \delta$), we would like to give $m_b^i$ preference over $m_a^i$. Therefore, we would like to set the hyperparameter $T$ so that the posterior of the path using $m_b^i$ is higher, i.e. $p(m_{<t}^1, ..., m_a^i | \mathbf{x}_1, ..., \mathbf{x}_N) < p(m_{<t}^1, ..., m_b^i | \mathbf{x}_1, ..., \mathbf{x}_N)$. Using Eq. 4 we can express this as:

$$p(m_{<t}^1, ..., m_a^i | \mathbf{x}_1, ..., \mathbf{x}_N) < p(m_{<t}^1, ..., m_b^i | \mathbf{x}_1, ..., \mathbf{x}_N)$$

$$p(m_a^i) p(D^{H^{i-1}} | m_a^i) < p(m_b^i) p(D^{H^{i-1}} | m_b^i)$$

$$\frac{p(m_a^i)}{p(m_b^i)} < \frac{p(D^{H^{i-1}} | m_b^i)}{p(D^{H^{i-1}} | m_a^i)}$$

$$\frac{e^{\frac{Acc(a)}{T}}}{e^{\frac{Acc(a)+\delta}{T}}} < \frac{p(D^{H^{i-1}} | m_b^i)}{p(D^{H^{i-1}} | m_a^i)} \tag{13}$$

$$\frac{Acc(a)}{T} - \frac{Acc(a) + \delta}{T} < \log \frac{p(D^{H^{i-1}} | m_b^i)}{p(D^{H^{i-1}} | m_a^i)}$$

$$T > \frac{\delta}{\log p(D^{H^{i-1}} | m_b^i) - \log p(D^{H^{i-1}} | m_a^i)}.$$

We can then use the inequality in Eq. 12 in order to determine the value T. To do this, one needs to decide how much difference in log likelihood should be overcome by a difference $\delta$ in accuracy.

## C  BELL - BENCHMARKS FOR LIFELONG LEARNING

We now introduce *BELL* - a suite of benchmarks for evaluating the aforementioned CL properties. As above, we assume compositional tasks and then generate various lifelong learning sequences, with each evaluating one or two of the desired properties. Running an CL algorithm on all sequences then allows us to asses which properties are present and which are missing. This builds

| Sequence | Sequence Pattern | Properties |
|---|---|---|
| $S^{pl}$ | $[\Psi_1^+, \Psi_2^+, \Psi_3^+, \Psi_4^+, \Psi_5^+, \Psi_6^+]$ | 1., 2. |
| $S^-$ | $[\Psi_1^+, \Psi_2^-, \Psi_3^-, \Psi_4^-, \Psi_5^-, \Psi_1^-]$ | 3. |
| $S^{out}$ | $[\Psi_1^+, \Psi_2^-, \Psi_3^-, \Psi_4^-, \Psi_5^-, \Psi_6^- = (D_1, h_1, g_6)]$ | 3.1 |
| $S^{out*}$ | $[\Psi_1^-, \Psi_2^+ = (D_1, h_1, g_2), \Psi_3^-, \Psi_4^-, \Psi_5^-, \Psi_6^- = (D_1, h_1, g_6)]$ | 3.1 |
| $S^{out**}$ | $[\Psi_1^-, \Psi_2^+ = (D_1, h_1, g_2), \Psi_3^-, \Psi_4^-, \Psi_5^-, \Psi_1^-]$ | 3.1 |
| $S^{in}$ | $[\Psi_1^+, \Psi_2^-, \Psi_3^-, \Psi_4^-, \Psi_5^-, \Psi_6^- = (D_6, h_6, g_1)]$ | 3.2.1 |
| $S^{sp}$ | $[\Psi_1^+, \Psi_2^-, \Psi_3^-, \Psi_4^-, \Psi_5^-, \Psi_6^- = (D_6, h_6, g_1)]$ | 3.2.2 |
| $S^{few}$ | $[\Psi_1^+ = (D_1, h_1), \Psi_2^+ = (D_2, h_2), \Psi_3^-, \Psi_4^- = (D_1, h_1, g_4), \Psi_5^-, \Psi_6^{--} = (D_2, h_2, g_4)]$ | 3.3 |
| $S^+$ | $[\Psi_1^-, \Psi_2^-, \Psi_3^-, \Psi_4^-, \Psi_5^-, \Psi_1^+]$ | 4. |
| $S^{long}$ | $[\Psi_i]_{i=1}^{100}$ | 5. |

Table 4: A list of all of the different CL sequences in *BELL*, each of which evaluates different CL properties. The first column contains the sequence's name, the second shows the sequence's pattern and the third column lists the CL properties evaluated by this sequence.

upon the CTrL benchmark suite, presented in Veniat et al. (2020), which defines different sequences of image classification tasks, namely $S^{pl}$, $S^-$, $S^{out}$, $S^{in}$, $S^+$ and $S^{long}$. They evaluate plasticity, perceptual transfer, non-perceptual transfer, catastrophic forgetting, backward transfer and scalability. We define these sequences similarly but for problems with compositional tasks. This allows us to introduce new sequences which evaluate new CL properties ($S^{sp}$ and $S^{few}$). We also introduce new more challenging sequences ($S^{out*}$ and $S^{out**}$).

We assume compositional tasks and represent each problem as a triple $\Psi_t = (D_j, h_j, g_k)$ where $D_j$ is the distribution of the inputs and $h_j$ and $g_k$ constitute the labelling function $f$, i.e. $f = g \circ h$. We refer to $h_j$ as the lower labelling sub-function and to $g_k$ as the upper labelling sub-function. We use the indices $j$ and $k$ to indicate whether the corresponding labelling sub-function has occurred before in the sequence ($j < t, k < t$) or if it is new and randomly selected ($j = t, k = t$). For brevity, if $j = t$ and $k = t$, we don't write out the whole triple but only $\Psi_t$. By repeating previously encountered domains or labelling sub-functions, we can control what knowledge can be transferred in each of the define sequences. In turn, this allows us to evaluate different CL properties. We use $\Psi^+$ to indicate that the dataset generated for this problem is sufficient to learn a well generalising approximation without transferring knowledge. On the other hand, $\Psi^-$ indicates that the CL algorithm cannot achieve good generalisation on this problem without transferring knowledge. Finally, $\Psi^{--}$ indicates that the generated training dataset consists of only a few datapoints, e.g. 10.

A complete list of the different sequences in *BELL* is presented in Table 4. Following Veniat et al. (2020), we set the sequence length of most sequences to 6 which, as we show in the experiments section, is sufficient for evaluating different CL properties. Next, we separately present each sequence, detailing which CL properties it evaluates.

**Plasticity** and **Stability**: The sequence $S^{pl} = [\Psi_1^+, \Psi_2^+, \Psi_3^+, \Psi_4^+, \Psi_5^+, \Psi_6^+]$ consists of 6 distinct problems, each of which has a different input domain and a different task. Moreover, each of the generated datasets has a sufficient number of data points as not to necessitate transfer. Therefore, this sequence evaluates a CL algorithm's ability to learn distinct problems, i.e. its plasticity (1.). Moreover, this sequence can be used to evaluate an algorithm's stability (2.) by assessing its performance after training on all problems and checking for forgetting.

**Forward Transfer**: Most of our sequences are dedicated to evaluate different types of forward transfer. To begin with, in the sequence $S^- = [\Psi_1^+, \Psi_2^-, \Psi_3^-, \Psi_4^-, \Psi_5^-, \Psi_1^-]$ the first and the last datasets represent the same problem, however, the last dataset has fewer data points. Therefore, an CL algorithm would need to transfer the knowledge acquired from solving the first problem, thus, demonstrating its ability to perform overall forward transfer (3.).

**Perceptual Forward Transfer**: We introduce three different sequences for evaluating perceptual transfer (3.1). First, in $S^{out} = [\Psi_1^+, \Psi_2^-, \Psi_3^-, \Psi_4^-, \Psi_5^-, \Psi_6^- = (D_1, h_1, g_6)]$ the last problem has the same input domain and input-processing target function $h_1$ as in problem 1. However, the last problem's dataset is small, therefore, an CL algorithm needs to perform perceptual transfer from the first problem, which is described by a large dataset. Second, $S^{out*} = [\Psi_1^-, \Psi_2^+ = (D_1, h_1, g_2), \Psi_3^-, \Psi_4^-, \Psi_5^-, \Psi_6^- = (D_1, h_1, g_6)]$ shares the same input distributions and lower la-

belling sub-function $h_1$ across problems $\Psi_1$, $\Psi_2$ and $\Psi_6$. Therefore, a CL algorithm needs to decide whether to transfer knowledge obtained from the first or from the second problem. Third, the sequence $S^{out**} = [\Psi_1^-, \Psi_2^+ = (D_1, h_1, g_2), \Psi_3^-, \Psi_4^-, \Psi_5^-, \Psi_1^-]$ is similar to the preceding one, with the distinction that the last problem is the same as the first. In this sequence, an CL algorithm needs to decide between reusing knowledge acquired from solving the same problem ($\Psi_1$), or to transfer perceptual knowledge from a more different problem ($\Psi_2$). Overall, these three sequences are designed to be increasingly more challenging in order to distinguish between different CL algorithms which are capable of perceptual transfer to a different extent.

**Non-Perceptual Forward Transfer**: Currently, we define two sequences to assess an algorithm's ability to transfer non-perceptual knowledge. Firstly, in $S^{in} = [\Psi_1^+, \Psi_2^-, \Psi_3^-, \Psi_4^-, \Psi_5^-, \Psi_6^- = (D_6, h_6, g_1)]$ the last problem has the same upper labelling sub-function as the first problem. However, the two problems' input distributions and lower labelling sub-functions are different. Therefore, an CL algorithm would need to transfer knowledge across different input domains (3.2.1). Secondly, the sequence $S^{sp} = [\Psi_1^+, \Psi_2^-, \Psi_3^-, \Psi_4^-, \Psi_5^-, \Psi_6^- = (D_6, h_6, g_1)]$ is simiarly defined, however, the input distribution of the last problem is also defined on a different input space from the input space of the first problem. Therefore, an algorithm would need to transfer knowledge across different input spaces (3.2.2).

**Few-shot Forward Transfer:** In order to evaluate this property, we introduce the following sequence, in which the first two problems are different from the rest of the sequences: $S^{few} = [\Psi_1^+ = (D_1, h_1), \Psi_2^+ = (D_2, h_2), \Psi_3^-, \Psi_4^- = (D_1, h_1, g_4), \Psi_5^-, \Psi_6^{--} = (D_2, h_2, g_4)]$. The labelling functions of the first two problems are simpler, each consisting only of a lower labelling sub-function. This is done in order to provide a CL algorithm with more supervision on how to approximate $h_1$ and $g_1$ more accurately. The fourth problem $\Psi_4$ in this sequence then shares the same input domain and lower labeling sub-function as the first problem, but introduces a new upper labelling sub-function $g_4$. The last problem then shares the input domain and the lower labelling sub-function of $\Psi_2$, while also sharing the upper labelling sub-function of problem $\Psi_4$. Moreover, the last problem's training dataset consists of only a few data points. Therefore, a CL algorithm would need to reuse its approximations of $h_2$ and $g_4$ in a novel manner in order to solve the last problem.

**Backward Transfer:** The sequence $S^+ = [\Psi_1^-, \Psi_2^-, \Psi_3^-, \Psi_4^-, \Psi_5^-, \Psi_1^+]$ has the same first and last problem. However, the first dataset has significantly less data points than the last. Ideally, an CL algorithm should use the knowledge acquired after solving the last problem in order to improve its performance on the first problem. While this sequence represents a starting point for evaluating backward transfer, it is possible to introduce other sequences, representing more elaborate evaluations. For instance, introducing sequences which evaluate perceptual and non-perceptual backward transfer separately. However, as backward transfer is not the focus of this thesis, this is left for future work.

**Scalability:** This property can be evaluated using a long sequence of problems. For this purpose we define $S^{long} = [\Psi_i]_{i=1}^{60}$ which consists of 60 problems, each randomly selected with replacement from a set of problems. Most problems are represented by a small dataset, $\Psi_t^-$. Each of the first 50 problems has a $\frac{1}{3}$ probability of being represented by a large dataset, $\Psi_t^+$. Each problem also has a $\frac{1}{10}$ probability of being represented by an extra small dataset, $\Psi_t^{--}$.

The definitions of these sequences rely on three sets of domains, lower labeling sub-functions and upper labelling sub-functions, respectively. In turn, these can be used to create a set of problems. Next, we present a set of problems which can be used together with the aforementioned sequence definitions in order to evaluate CL algorithms.

## C.1 COMPOSITIONAL PROBLEMS

To implement the sequences defined above, one needs to define a set of compositional problems. To this end, we define 9 different pairs of an input domain and a lower labelling sub-function, $\{(D_i, h_i)\}_{i=1}^9$. Moreover, we define 16 different upper labelling sub-functions $\{g_i\}_{i=1}^{16}$. These can be combined into a total of 144 different compositional problems.

First, we define 9 image multi-class classification tasks, which all share input and output spaces $\mathbb{R}^{28 \times 28} \to \mathbb{R}^8$, but each have a different input distribution $D_i$ and a domain-specific labelling function $h_i$. Concretely, we start with the following image classification datasets: MNIST (LeCun et al., 2010), Fashion MNIST (FMNIST) (Xiao et al., 2017), EMNIST (Cohen et al., 2017) and Kuzushiji49 (KMNIST) (Clanuwat et al., 2018). Since some of the classes in KMNIST have

significantly fewer training data points, we only use the 33 classes with the following indices: $[0, 1, 2, 4 - 12, 15, 17 - 21, 24 - 28, 30, 34, 35, 37 - 41, 46, 47]$, as they have a sufficient number of associated data points. We split the image datasets into smaller 8-class classification datasets. We use $i$ to denote the different splits of the same original dataset. For instance EMNIST$_2$ represents the third split of EMNIST, corresponding to a classification task among the letters form 'i' to 'p'. Using this we end up with the following 9 image datasets: MNIST$_1$, FMNIST$_1$, $\{$EMNIST$_i\}_{i=1}^3$, $\{$KMNIST$_i\}_{i=1}^4$. For each of these image datasets, we set aside 4800 validation images from the training dataset. We also keep the provided test images separate.

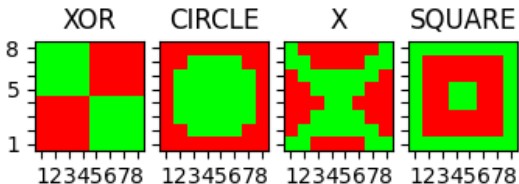

Figure 3: An illustration of the four two-dimensional patterns which are used by the four $g^{(2)}$ functions to label the input coordinates. Green indicates a positive label, and red indicates a negative label.

Second, we define a set of binary classification tasks, which map $\mathbb{R}^{16} \to \{0, 1\}$. Each task's labelling function $g_i$ receives two concatenated 8-dimensional one-hot encodings and returns a binary value, indicating if the given combination of 2 classes, represented by the input, fulfils a certain criteria. We further decompose the labelling function into $g_i(\mathbf{x}) = g_k^{(2)}(g_j^{(1)}(\mathbf{x}[: 8]), g_j^{(1)}(\mathbf{x}[8 :]))$. Here, $g_j^{(1)}$ maps a one-hot encoding to an integer between 1 and 8. For instance, $g_1^{(1)}$ maps the first dimension to 1, the second to 2 and so on. As a result, we use $g^{(1)}$ to convert the initial input of two one-hot encodings to two-dimensional coordinates. We define 4 different $g^{(1)}$ mappings, where $g_1^{(1)}$ is defined as above, and $g_1^{(2)}$, $g_1^{(3)}$ and $g_1^{(4)}$ each map the dimensions to a different randomly selected integer between 1 and 8.

At the same time, each $g_k^{(2)} : \mathbb{R}^2 \to \{0, 1\}$ outputs whether a given two-dimensional coordinate is a part of a certain pattern or not. We define 4 different $g^{(2)}$ functions, each corresponding to one of 4 two-dimensional patterns, shown in Fig 3. In total, these functions need to label $8 * 8 = 64$ different two-dimensional coordinates.

We fuse the 4 different $g^{(1)}$ functions with the 4 different $g^{(2)}$ functions to define 16 different $g$ functions:

$$\{g_{(k-1)*4+j}(\mathbf{x}) = g_k^{(2)}(g_j^{(1)}(\mathbf{x}[: 8]), g_j^{(1)}(\mathbf{x}[8 :])), k \in \{1, 2, 3, 4\}, j \in \{1, 2, 3, 4\}\}.$$

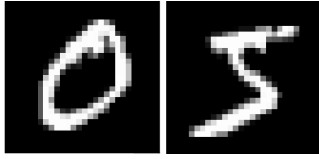

Figure 4: An example input for $\Psi = (D_{\text{MNIST}_1}, h_{\text{MNIST}_1}, g = (g_{\text{XOR}}^{(2)}, g_1^{(1)}))$. These images are classified by $h_{\text{MNIST}}$ and then are mapped to the coordinates (1, 6) by $g_1^{(1)}$ since they represent the first and sixth classes respectively. Afterwards, $g_{\text{XOR}}^{(2)}$ labels this input as 0, using the XOR pattern, shown in 3.

Finally, we can combine our 9 image classification datasets $\{(D_i, h_i)\}_{i=1}^9$ with our 16 binary classification tasks, in order to create 144 compositional problems $\{\Psi_{(k-1)*9+j} = (D_k, h_k, g_j), k \in \{1, ..., 9\}, j \in \{1, ..., 16\}\}$. The input to a problem $\Psi_i = (D_k, h_k, g_j)$ are two images sampled from $D_i$. Each image is labelled by $h_i$, each resulting in an eight-dimensional one-hot encoding of the corresponding image's class. The two one-hot encodings are then concatenated and labelled by $g_j$,

which results in a binary label. An example for $\Psi = (D_{\mathrm{MNIST}_1}, h_{\mathrm{MNIST}_1}, g = (g_{\mathrm{XOR}}^{(2)}, g_1^{(1)}))$ is shown in Fig 4.

Sequence $S^{\mathrm{sp}}$ involves transferring across input spaces by having its last problem's input domain be defined over a different input space. To create this domain we flatten any randomly selected domain from $\mathbb{R}^{28 \times 28}$ to $\mathbb{R}^{784}$. This loses the images' spacial information and requires that a different neural architecture is applied to process those inputs.

## C.2 REALISING THE SEQUENCES

To implement a sequence S of length $l$, we need to select $l$ concrete compositional problems which fit the pattern specified by said sequence. Let the sequence have $l^{(1)}$ different pairs of image domain and lower labelling sub-function, and $l^{(2)}$ different upper labelling sub-functions. For all sequences, apart from $S^{\mathrm{long}}$, we select $l^{(1)}$ pairs of $(D_i, h_i)$ by sampling from the set of all possible image classification tasks, without replacement. Similarly, we select $l^{(2)}$ different upper labelling sub-functions by sampling without replacement from the set of available binary classification tasks $\{g_i\}_{i=1}^{16}$. For $S^{\mathrm{long}}$, we use sampling with replacement.

If a problem's training dataset needs to be large, $\Psi_i^+$, we generate it according to the triple $n_{\mathrm{tr}}^+ = (30000, \mathrm{All}_{\mathrm{tr}}, \mathrm{All})$. The first value indicates that we generate 30000 data points in total. The second value indicates how many unique images from the ones set aside for training, are used when generating the inputs. In this case, we use all the available training images. The third value indicates how many out of the 64 unique two-dimensional coordinates, used by the upper labelling sub-function, are represented by the input images. In this case, we use all two-dimensional coordinates.

Some of the problems' training datasets are required to be small and to necessitate transfer. For sequences $S^-, S^{\mathrm{out}}, S^{\mathrm{out*}}, S^{\mathrm{out**}}, S^{\mathrm{few}}, S^+$, we generate the training datasets of each problem $\Psi^-$ using the triple $n_{\mathrm{tr}}^- = (10000, 100, \mathrm{All})$. This way, only 100 unique images are used to generate the training dataset, so solving the problem is likely to be difficult without perceptual transfer. The subset of unique images is randomly sampled and can is different between two problems which share an input domain. For sequences $S^{\mathrm{in}}$ and $S^{\mathrm{sp}}$, which evaluate non-perceptual transfer, we use the triple $n_{\mathrm{tr}}^- = (10000, \mathrm{All}_{\mathrm{tr}}, 30)$. As a result, the generated datasets will only represent 30/64 of the two-dimensional coordinates, which is not sufficient for learning the underlying two-dimensional pattern. Therefore, these problems will necessitate non-perceptual transfer. When generating a dataset for a problem $\Psi^-$ in the sequence $S^{\mathrm{long}}$, we randomly choose between the two, namely between $(10000, 100, \mathrm{All})$ and $(10000, \mathrm{All}_{\mathrm{tr}}, 30)$.

For the problems in which the training dataset needs to contain only a few data points, $\Psi^{--}$, we use the triple $n_{\mathrm{tr}}^{--} = (10, 20, 10)$. This creates only 10 data points, representing 20 different images and 10 different two-dimensional patterns.

For problems with $\Psi^{--}$, we use the triple $n_{\mathrm{val}}^{--} = (10, 20, 10)$ for generating the validation dataset. For the rest of the problems, we use the triple $n_{\mathrm{val}}^{--} = (5000, \mathrm{All}_{\mathrm{val}}, \mathrm{All})$. Finally, we generate all test datasets using the triple $n_{\mathrm{test}}^{--} = (5000, \mathrm{All}_{\mathrm{test}}, \mathrm{All})$.

## D EXPERIMENTS

Across both benchmark suites, we use the same set of hyperparameters for our algorithms, which we initially selected for a different set of problems in our preliminary experiments. This suggests that this choice of hyperparameters is robust and applicable to different problems and neural architectures. For PeCL and CCL we project the hidden states to 20 dimensions before modelling the resulting distribution. Moreover, we use 0.001 as the softmax temperature parameter for the prior distribution over pre-trained modules. For NoCL and CCL we store 40 of the training inputs of each of the pre-trained modules used in the $(L - l_{\min} + 1)$th layer. The value for $l_{\min}$ is 3 for both the BELL and the CTrL benchmarks. The jitter of the expected improvement is 0.001 and the threshold used for early stopping, is 0.001. To calculate the value of the upper confidence bound (UCB), we set the hyperparameter $\beta = 2$ in order to encourage exploration over exploitation.

We use two measurements to assess the performance of each LML algorithm. First, we compute the average accuracy, $\hat{\mathcal{A}}$, across all problems after the last problem is solved. An algorithm's average accuracy after it is trained on some problem sequence $S = (\Psi_1, \Psi_2, ..., \Psi_{|S|})$ can be defined as:

$$\hat{\mathcal{A}}(S) = \frac{1}{|S|} \sum_{i=1}^{|S|} \mathcal{A}(\Psi_i) \tag{14}$$

where $\mathcal{A}(\Psi_i)$ denotes the algorithms' final accuracy on problem $\Psi_i$, evaluated on a held-out test dataset. Second, we compute the forward transfer on the last problem only, $Tr^{-1}$, for the sequences in which the performance on the last problem diagnoses an LML property. We compute it as the difference between the final accuracy on the last problem by an LML algorithm, and this accuracy for a standalone baseline, $\mathcal{A}_{\text{SA}}$:

$$Tr^{-1}(S) = \mathcal{A}(\Psi_{|S|}) - \mathcal{A}_{\text{SA}}(\Psi_{|S|}). \tag{15}$$

We do not evaluate backward transfer, nor forgetting, since all the LML algorithms which we evaluate are immune to forgetting, and to backward transfer, by design as they freeze any previously trained parameters.

In our experiments, we make the training process deterministic, so that the difference in performance can be accredited only to the LML algorithm, and not due to randomness introduced during training. For this purpose, we fix the random initialisation of new parameters to be problem and path-specific. In other words, for a given problem, if a model with the same path is instantiated twice, it will have the same initial values for its new randomly initialised parameters. Moreover, we fix the sequence of randomly selected mini batches seen during training to be the same for a given problem. Finally, as we use PyTorch for our experiments, we fix the random seed and use the command "torch.use_deterministic_algorithms(True)". The overall results is that, for a given problem and a given library, evaluating the same path will always result in the same performance, even across different modular LML algorithms.

All experiments are implemented using PyTorch 1.11.0 (Paszke et al., 2019). We also use GPy's (GPy, since 2012) implementation of a Gaussian process. We run each LML algorithm on a single sequence, on a separate GPU. All experiments are run on a single machine with two Tesla P100 GPUs with 16 GB VRAM, 64-core CPU of the following model: "Intel(R) Xeon(R) Gold 5218 CPU @ 2.30GHz", and 377 GB RAM.

## D.1 BELL BENCHMARKS

We first evaluate our algorithms and the baselines on the benchmark suite which we introduced in Section C. We create 3 versions of each sequence by randomly selecting different compositional problems. Then, for each sequence, we report the measurements, averaged over these 3 versions.

### D.1.1 NEURAL ARCHITECTURE

Here, we present the minimal neural architecture which we have found to be suitable for solving said problems.

We first define a convolutional neural network $\zeta_{\text{CNN}} : \mathbb{R}^{28 \times 28} \to \mathbb{R}^8$, suitable for processing images from the image classification datasets. We use a 5-layer architecture with *ReLU* hidden activations and a *softmax* output activation. The layers are as follows: *Conv2d(input_channels=1, output_channels=64, kernel_size=5, stride=2, padding=0)*, *Conv2d(input_channels=64, output_channels=64, kernel_size=5, stride=2, padding=0)*, *flatten*, *FC(4\*4\*64, 64)*, *FC(64, 64)*, *FC(64, 10)*. Here, *Conv2d* specified a two-dimensional convolutional layer and *FC* specifies a fully-connected layer.

Second, we define a fully-connected neural network for processing a concatenation of two 8-dimensional one-hot embeddings, $\zeta_{\text{MLP}} : \mathbb{R}^{16} \to \mathbb{R}^1$. It consists of 2 *FC* hidden layers with $64$ hidden units and *RELU* hidden activations, followed by an output *FC* layer with a *sigmoid* activation.

For a compositional problem $\Psi_k = (D_i, h_i, g_j)$ the input is a 2-tuple of images, $(\mathbf{x}^1, \mathbf{x}^2)$ and the expected output is a binary classification. We solve it using the architecture $\zeta_{\text{comp}} =$

$\zeta_{\text{MLP}}(concatenate(\zeta_{\text{CNN}}(\mathbf{x}^2), \zeta_{\text{CNN}}(\mathbf{x}^2)))$. This architecture processes each of the 2 input images with the same $\zeta_{\text{CNN}}$ model. Then the 2 outputs are concatenated and processed by a $\zeta_{\text{MLP}}$ model.

We represent this as a modular neural architecture by considering each of the 8 parameterised non-linear transformations to be a separate module. This increases the number of possible paths for each problem. As a result, for the 6th problem in a sequence, the number of possible paths is upper bounded by $\mathcal{O}(6^8 = 1679616)$. Therefore, in this setting, even sequences of length 6 are challenging for modular LML approaches.

The input space of the last problem of sequence $S^{\text{sp'}}$ is given by a 8-dimensional vector. Therefore, only for this problem, we replace $\zeta_{\text{CNN}}$ with a different architecture, $\zeta_{\text{FC}}$, which consists of two fully connected layers, with a hidden size of 64, uses $ReLU$ as a hidden activation and *softmax* as its output activation.

We train new parameters to increase the log likelihood of the labels using the AdamW optimiser (Loshchilov & Hutter, 2017) with 0.00016 learning rate, and 0.97 weight decay. The training is done with a mini batch size of 32 and across 1200 epochs. We apply early stopping, based on the validation loss. We stop after 6000 updates without improvement and return the parameters which were logged to have had the best validation accuracy during training.

## D.2 CTRL BENCHMARKS

The CTrL benchmark suite was introduced in Veniat et al. (2020). They define a number of sequences, based on seven image classfication tasks, namely: CIFAR10 and CIFAR100 (Krizhevsky et al., 2009), DTD (Cimpoi et al., 2014), SVHN (Netzer et al., 2011), MNIST (LeCun et al., 1998), RainbowMNIST (Finn et al., 2019), and Fashion MNIST (Xiao et al., 2017). All images are rescaled to 32x32 pixels in the RGB color format. CTrL was first to introduce the following sequences: $S^-$, $S^+$, $S^{\text{in}}$, $S^{\text{out}}$, $S^{\text{pl}}$ and $S^{\text{long}}$, which are defined similarly to our definitions. However, the difference is that they are defined for and implemented by image classification tasks. The last task in $S^{\text{in}}$, which evaluates non-perceptual transfer, is given by MNIST images with a different background color than the first task. The last task in $S^{\text{out}}$ is given by shuffling the output labels of the first task. $S^{\text{long}}$ has 100 tasks. For each task, they sample a random image dataset and a random subset of 5 classes to classify. The number of training data points is sampled according to a distribution that makes it more likely for later tasks to have small training datasets. In contrast to us, they use only 1 selection of tasks for each sequence, i.e. 1 realisation of each sequence. To generate the sequences, we use the code provided by the authors (Veniat & Ranzato, 2021).

The neural architecture used is a small variant of ResNet18 architecture which is divided into 6 modules, each representing a different ResNet block (He et al., 2016). While the paper presenting the CTrL benchmark states that 7 modules are used, we used the authors' code (Veniat, 2021) for this method which specifies only 6 modules with the same total number of parameters. The difference from the architecture stated in the paper is that the output layer is placed in the last module, instead of in a separate module.

All parameters are trained to reduce the cross-entropy loss with an Adam optimiser Kingma & Ba (2014) with $\beta_1 = 0.9$, $\beta_1 = 0.999$ and $\epsilon = 10^{-8}$. For each task, each path is evaluated 6 times with different combinations of values for the hyperparameters of the learning rate ($\{10^{-2}, 10^{-3}\}$) and of the weight decay strength $\{0, 10^{-5}, 10^{-4}\}$. The hyperameters which lead to the best validation performance are selected. Early stopping is employed during training. If no improvement is achieved in 300 training iterations, the parameters with the the best logged validation performance are selected. Data augmentation is also used during training, namely random crops (4 pixels padding and 32x32 crops) and random horizontal reflection.

# E ABLATION EXPERIMENTS

## E.1 BAYESIAN OPTIMISATION

We evaluate our search strategy $\mathbb{S}_{\text{BO}}^{\text{NT},l}$ (Eq. 10) which is used for non-perceptual transfer. For this purpose, we create a new sequence:

$$S^{\text{in+}} = [\Psi_1^+, \Psi_2^+, ..., \Psi_{15}^+, \Psi_{16}^- = (D_6, h_6, g_1)]$$

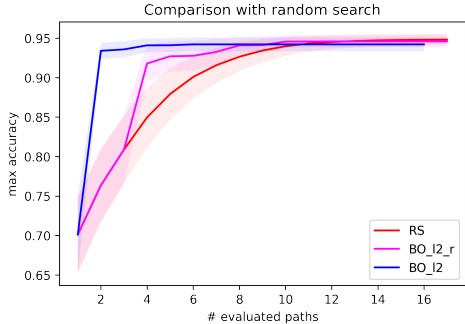

Figure 5: Comparing to randomly selected initial points (BO_l2_rnd_init), and random search (RS).

which involves all 16 upper labelling sub-functions $g$ of BELL. The last problem's dataset is generated according to the triple $n_{tr}^- = (10000, \text{All}_{tr}, 30)$, which states that only 30 out of the 64 possible two-dimensional patterns are represented in the dataset. As a result, non-perceptual transfer is necessary in order to maximise the performance on the final problem. We create 5 realisations of $S^{in+}$ with different randomly selected problems.

Even though our method is deterministic, the baselines we compare it to involve randomly selecting 2 or all random paths. For each such baseline, and each of the 5 realisations of the sequence $S^{in+}$, we run the baseline 10 times with 10 different random seeds. This results in $10 * 5 = 50$ evaluations per method which we average over when reporting its performance. For each method, we plot its maximum accuracy achieved per number of paths evaluated.

We compare our search strategy $s_{BO}^{NT,l}$ (which we denote BO_l2) to an augmented version which randomly selects the initial 2 paths (BO_l2_r), and to a random search baseline which recommends paths in a random order (RS). The results, shown in Fig. 5, demonstrate our approach's superior anytime performance.

### E.2 RANDOM PROJECTIONS

The usage of $\mathbb{S}_G^{PT}$ (Eq. 5) relies on approximations of the training input distributions of pre-trained modules. Instead of modelling a module's training input distribution directly, we proposed to first project samples from it to $k$ dimensions using random projection, and then we model the resulting distribution with a multivariate Gaussian. In this section we would like to evaluate three aspects of this approach. First, we would like to assess the usefulness of the resulting approximations for the purposes of selecting the correct input distribution. Second, we would like to assess the sensitivity of our approximations to the hyperparameter $k$. Third, we would like to compare our approach to Gaussian approximation of the original input space in order to determine whether we sacrifice performance.

To this end, we evaluate whether our approach is useful for distinguishing between a set of input distributions. We compare the approximations resulting from different choices of $k = \{\mathbf{10}, 20, 40\}$. The resulting methods are referred to as $rp\_10$, $rp\_20$ and $rp\_40$ respectively. Moreover, we compare to the method of computing a Gaussian approximation of a module's training input distribution, without a random projection. Since this can lead to a singular covariance matrix, we make use of diagonal loading (Draper & Smith, 1998) in which we add a small constant ($10^{-8}$) to the diagonal of the computed sample covariance matrix in order to make it positive definite. We refer to the resulting method as *diag_loading*.

We compare how well can these approaches distinguish between the 9 image datasets used in BELL. We chose to use the input images for our comparison since they have the highest dimension and should be the most difficult to approximate.

To evaluate one of the methods, we first use it to approximate all 9 input distributions using $N$ data points, resulting in 9 approximations, denoted as $\{q_i\}_i^9$. Second, for each input distribution $p_j$, we sample 100 different data points and use them to order the approximations in descending order of their likelihood. Ideally, if the data points are sampled from the $j$-th distribution $p_j$, the

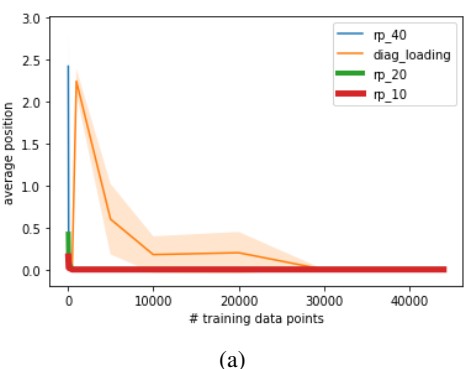
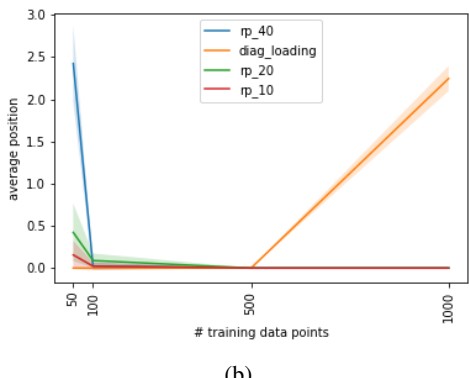

(a)                                              (b)

Figure 6: Comparison of different methods for modelling a module's input distribution. The x axis represents the number $N$ of data points used to compute an approximation. The y axis represents the average position, as defined in the main text, which indicates how well a method can approximate the distributions. The lower the average position is, the better the model performs. Figure a) presents a plot across all choices of $N$. Figure b) focuses on the first few values of $N$.

corresponding approximation $q_j$ should have the highest likelihood, and thus should be the first in the list, i.e. should have an index equal to $0$. We compute the index of $q_j$ in the ordered list and use it as an indication of how successfully the method has approximated $p_j$. We compute this index for each of the 9 distributions and report the average index, also referred to as the *average position*.

We evaluate each method for different choices of $N$, $N = \{50, 100, 500, 1000, 5000, 10000, 20000, 30000, 44000\}$. Moreover, we repeat all evaluations 5 times using different random seeds and report the mean and standard error of the average position. The results are reported in Fig. 6.

Our results show that directly modelling the original distribution with a Gaussian leads to sub-optimal performance. On the other hand, we observe that for $N \geq 500$, the methods which use random projection can always match the given data points with the correct distribution which they were sampled for. Surprisingly, for $N = 50$ and $N = 100$, diag_loading outperforms the other methods and can successfully identify the correct distribution of the given data points. Furthermore, we observe that for these values of $N$, decreasing the dimension $k$ that the data points are projected to leads to better performance of the methods that are based on random projection.

Overall, our results suggest that the approximations which we use for $\mathbb{S}_G^{\mathrm{PT}}$ are effective when the new modules are trained on more than $100$ data points. This seems like a reasonable requirement, as fewer points are likely to result in a sub-optimal performance.

