# OpenReview forum: "A Probabilistic Framework For Modular Continual Learning"
_ICLR.cc/2023/Conference — Submitted to ICLR 2023_

### Official Review · Reviewer_ifu1 · 2022-10-24

**Confidence:** 5
**Correctness:** 3
**Technical Novelty And Significance:** 3
**Empirical Novelty And Significance:** 3
**Recommendation:** 3

**Clarity, Quality, Novelty And Reproducibility:**

Clarity
=====
I found this work difficult to read and understand. Here are some unclear points I found while reviewing the submission:

* To my understanding, only one module is chosen per layer. Then, at the beginning of Section 3, you mention that you split $\Pi_t$ in subsets $\Pi_t^i$. Does that mean that you choose modules layerwise? If so, why do you call $\Pi_t^i$ a subset if it is a single module? Maybe I am missing something.
* In equation 3, what is the j subscript after the parenthesis?
* In equation 6, q is introduced without any definition. I could get it from the context but a more explicit definition would help. The same with $\mathbf{A}$
* In section 4.2: "we freeze this selection and reuse the same l modules for PT paths which transfer more modules" what do you mean?
* "We define the augmented search strategy" why augmented?
* It took some time for me to parse equation 7, I think some explanation or guidance to the reader (particularly about $\pi^{*PT,l-1}[l-1]$) could make it much easier to understand.
* "The difference between two NT paths ... of length l is that their last l pre-trained modules compute different functions." If they are of length l, the last l pre-trained modules is the whole path no?
* I understand equation 10 but how it is used in your algorithm is not clear to me.
* "number of transfer which need to be transferred in order to improve the performance." what do you mean by the transfer that needs to be transferred?
* Eq 11. What is the $j$ after the parenthesis?
* You introduce the term LCB but I could not find an explanation for what it is. The search brought me to the appendix, where I could find what the letters mean as well as many "??" for the references.

Besides these points there are some minor typos like "as it does not does not involve" in Section 3.

Quality
=====
* For the most part, the technical quality is good, however, the lack of clarity hinders my understanding of the method. In experiments,  it is not clear why the authors did not compare to LMC.
* It is not clear whether the number of parameters of their method to put them in comparison to the other baselines (if it is the same I did not find it clearly stated in the text, maybe appendix C.2 last paragraph?)

Novelty
======
The proposed method is novel to the best of my knowledge.

Reproducibility
===========
The authors did not provide the code neither a reproducibility statement, however, the appendix contain some implementation details.

**Strength And Weaknesses:**

Strengths
=======
* The method is sound and it tries to approach modularity from first principles with a divide-and-conquer approach
* The proposed method outperforms MNTDP

Weaknesses
=========
* The model description is not clear, which makes it difficult to understand. Overall the submission looks unpolished. I highly encourage the authors to carefuly rework sections 3, 4, and 5, to make it clearer.
* The authors cite LMC but do not compare with it, is there any reason?
* The divide-and-conquer approach has a clear disadvantage: it introduces the additional complexity of establishing where to split the preceptual part from the non-perceptual part. Maybe the authors could include a limitations section to talk about that.

**Summary Of The Paper:**

The authors propose a probabilistic framework to model modular architectures for the problem of continual learning (CL). They divide the problem of choosing modules in two parts: perceptual transfer (PT), and non-perceptual transfer (NT). In PT, the first l layers are assumed to be pre-trained and the model must choose between those modules to maximize performance when training new modules on top. NT works the other way around. From this, they derive 3 CL algorithms: (i) PeCL, which does perceptual and few-shot transfer, (ii) NoCL, which does non-perceptual transfer, and (iii) CCL, which combines both strategies. To evaluate their model, the authors introduce a compositional version of CTrL and evaluate plasticity, backward transfer, forward transfer, perceptual transfer, few-shot transfer, and non-perceptual transfer. In experiments they show that their models perform favourably compared to SA, MCL-RS, HOUDINI, MNTDP.

**Summary Of The Review:**

The proposed method is sound and it tackles an interesting problem for continual learning. However, the lack of clarity of the text makes it difficult to properly asses the quality of this work. In its current form, this work does not meet the standards of ICLR.

---

> ### Author Response · Authors · 2022-11-21
> **Response**
>
> Thank you for your review.
>
> w1: We have updated our presentation to address your concerns regarding the clarity of the paper.
>
> w2: Please refer to our main comment about a comparison between our method and LMC.
>
> w3: Our divide-and-conquer approach hinges on predetermined subsets of paths which are not problem-specific. The difference from previous work [1], [2] is that we are explicit about the subsets of paths which we explore. In addition, our method explores a larger variety of module compositions which allows us to achieve all three forward transfer properties, in contrast to previous work.
>
> Quality: Please refer to our main comment regarding a comparison between our method and LMC [2].
> The neural architecture in our experimental setup is the same across all competing methods, leading to the same number of neural parameters. We do not account for additional parameters from our method because our approximations are negligible, requiring 440 parameters for an approximation.
>
> In terms of reproducibility, we will make our code available upon publication.
>
>
> [1] Veniat, T., Denoyer, L. and Ranzato, M.A., Efficient Continual Learning with Modular Networks and Task-Driven Priors.
> [2] Ostapenko, O., Rodriguez, P., Caccia, M. and Charlin, L., 2021. Continual learning via local module composition. Advances in Neural Information Processing Systems, 34, pp.30298-30312.

---

### Official Review · Reviewer_LPZE · 2022-10-25

**Confidence:** 4
**Correctness:** 2
**Technical Novelty And Significance:** 2
**Empirical Novelty And Significance:** 2
**Recommendation:** 3

**Clarity, Quality, Novelty And Reproducibility:**

In my view this work has potential but is work in progress, and is not quite ready for publication.

**Strength And Weaknesses:**

## Strengths

**S1.** The paper identifies limitations of prior work and seeks to address them.

## Weaknesses

**W1.** The contributions of the submission are very incremental. I see it as an extension of the work of (Venital et al. 2020) in the two directions outlined in the summary. Further, the theoretical contributions are obfuscated by the rough presentation (more on this below) and the validation is, in part, designed for the proposed method (i.e., some of the evaluation sequences are designed to not be handled properly by prior work).

**W2.** The presentation has many issues:

1. often symbols are introduced without definition and sometimes never get defined, e.g., Z in eq. (10), d in eq. (10) does not match the d previously defined; A at the end of page 4;

2. evaluation metrics like those in Table 1 are never defined; etc.

3. some explanations do not make much sense, e.g., “gives preference to modules which helped achieve a higher accuracy, on the problems which they have been trained on,” “minimum number of transfer which need to be transferred,” “can be explored by applying this search strategy to each sequentially,” etc.

4. the probabilistic models proposed are not explained in detail, e.g., eq. (5) is simply stated with no explanation and the Gaussian Process in section 5.1 is not even specified.

**W3.** The paper disregards other benchmarks in CL like (Lin et al. NeurIPS 2021.) Such a benchmark should be discussed, and results on it should either be included or there should be a discussion explaining why those would not be relevant.

## References

Lin et al. The CLEAR Benchmark: Continual LEArning on Real-World Imagery. NeurIPS 2021.


**Summary Of The Paper:**

This paper extends prior work on Continual Learning in two directions: 1) It introduces new learning sequences designed to evaluate non-perceptual transfer (transfer the last l layers) and few-shot transfer (transfer all layers) as well as more challenging sequences for the evaluation of perpetual transfer (transfer the first l layers). 2) Building on modular approaches to CL, e.g., (Venital et al. 2020), a probabilistic framework is proposed to tackle the exponential space of module combinations. One probabilistic model is proposed for perceptual (and few-shot) transfer and a separate model is proposed for non-perceptual transfer.

Experiments compare the proposed approach to some recent methods and the results are somewhat favorable to the proposed method.


**Summary Of The Review:**

The paper’s contribution is limited as outlined above and the quality of the presentation is very poor.

---

> ### Author Response · Authors · 2022-11-21
> **Response**
>
> W1: Thank you for your feedback. We revised the paper's introduction to make the novel technical contributions clearer. In summary, there are two main differences between our approach and MNDP-D [1]. First, we introduce a probabilistic approach which allows us to select novel input-dependent compositions of pre-trained modules. In contrast, MNTDP-D uses past solutions as feature extractors in order to select the solution closest to the new problem, and then tries to reuse a different number of the first layers from the selected solution. This strategy has two issues: (i) few-shot transfer is hard, and (ii) it is difficult to find the correct modules to transfer when only the first few modules are relevant to the new problem. As you have pointed out, we designed sequences $S^{few}$ and $S^{out**}$ in order to exemplify these shortcomings.
>
> Second, supporting non-perceptual transfer requires significant technical developments that did not appear in the previous work (e.g., the Gaussian process model in Section 5). The fundamental reason for this is that non-perceptual transfer is harder than perceptual transfer. For perceptual transfer, we can evaluate the lower layers from a previous task on the current dataset --- this gives a lot of information about whether the network is useful. For non-perceptual transfer, we cannot do this, because the lower layers are randomly initialized and contain no information. So estimating the performance of a path is much more difficult.
>
> W2: Thank you for the useful list of presentation issues. As mentioned in our main comment, we have updated our presentation in order to address your concerns.
>
> W3: We chose to evaluate our method on benchmarks which can evaluate different forward transfer properties. We selected two benchmark suites, including CTrL [1] which is the only benchmark suite which our closest competitors’ [1, 2] experiments are based on.
>
>
> [1] Veniat, T., Denoyer, L. and Ranzato, M.A., Efficient Continual Learning with Modular Networks and Task-Driven Priors.
> [2] Ostapenko, O., Rodriguez, P., Caccia, M. and Charlin, L., 2021. Continual learning via local module composition. Advances in Neural Information Processing Systems, 34, pp.30298-30312.

---

### Official Review · Reviewer_ARaq · 2022-10-25

**Confidence:** 4
**Correctness:** 3
**Technical Novelty And Significance:** 3
**Empirical Novelty And Significance:** 3
**Recommendation:** 5

**Clarity, Quality, Novelty And Reproducibility:**

Clarity: Different sections are well written.

Quality: The paper tackles an important problem.

Novelty: The underlying idea seems interesting, and to the best of my knowledge has not been explored in this context before.



**Strength And Weaknesses:**

Strengths:

-  The paper tackles an important problem i.e., exploring better ways of searching through possible combinations of modules.

Weaknesses:

- It would be useful to study how the proposed method scales as a function of number of modules.
- It would also be useful how the proposed method compares to methods which does a "soft" selection of the output of modules (like in LMC).
- The paper seems a bit hard to parse. It may be useful to give a high level "overview"of the proposed method before jumping to describe how the paper works.


**Summary Of The Paper:**

The paper tackles the idea of doing inference over a set of modules such that the resulting model can generalize as well as adapt quickly. Different approaches differ as to how they do search over an ensemble of modules. The proposed paper deals with both "perceptual" transfer and "non-perceptual" transfer. They introduce a probabilistic framework for selection of pre-trained modules. The key idea for driving the selection (i.e., which module to select) is  how well the module can transform the input. Each module is parameterized as a neural network. The paper define a posterior and prior, where the posterior is a function of how well the module can explain the input, and prior gives preference to modules which help achieve a higher accuracy.

**Summary Of The Review:**

The paper proposes a probabilistic framework to do inference over an ensemble of modules.

---

> ### Author Response · Authors · 2022-11-21
> **Response**
>
> Thank you for your review. We have updated the paper to improve its presentation. Please refer to the main comment for a detailed discussion on the differences between our method and LMC.

---

### Official Review · Reviewer_E5fq · 2022-10-26

**Confidence:** 4
**Clarity, Quality, Novelty And Reproducibility:** great on all accounts.
**Correctness:** 4
**Technical Novelty And Significance:** 4
**Empirical Novelty And Significance:** 2
**Recommendation:** 6

**Strength And Weaknesses:**

**Strengths**

The proposed method is sensible and adequately scales w.r.t. the number of training modules, which was a limitation of [1].
This work can inspire future work in continual learning.

**Weaknesses**

I wish the paper offered a more profound contrast with [1], both technically and empirically.

The idea of using the modules' input distribution as a signal for activating the modules was proposed in [1], but section 4 does not acknowledge that. Section 4 should explain how previous work has solved the problem above and then explain how their Bayesian treatment relates. The Section should further explain the pros and cons of such treatment compared to the previous solution.

The paper could instead (or additionally) benchmark their method against [1], which is the most relevant baseline for the paper.
I understand that running new experiments during rebuttal is not ideal.
I'm not expecting the authors to do that.
At least the others might want to add the reported results of [1] for the CTRL benchmark while mentioning that [1] operates in a much more challenging setting, i.e., **class**-incremental learning and not task-incremental.



**Summary Of The Paper:**

This paper introduces a probabilistic framework for modular continual learning.
They built on the idea in [1] that modules can be chosen locally based on their input distribution.
Contrarily to [1], this paper opts for a Bayesian treatment of the module composition problem.

The paper proposes two transfer learning settings for compositionality: perceptual vs non-perceptual transfer.
In the former, we assume that most of the drift in p(x,y) happens in p(x), whereas the in the latter, it occurs in p(y|x).
They prescribe one probabilistic method for both settings.

Some experiments in CTRL [2] and a modified version show that the method can outperform some baselines.

[1] Continual Learning via Local Module Composition. Oleksiy Ostapenko, Pau Rodríguez, Massimo Caccia, Laurent Charlin
[2] Efficient Continual Learning with Modular Networks and Task-Driven Priors. Tom Veniat, Ludovic Denoyer, Marc'Aurelio Ranzato

**Summary Of The Review:**

I think the paper is worthy of a publication if the weakness mentioned above are addressed.

---

> ### Author Response · Authors · 2022-11-21
> **Response**
>
> Thank you for your review. We have amended Section 4, following your recommendation. In addition, please refer to our main comment regarding a more detailed comparison between our method and LMC.

---

### Author Response · Authors · 2022-11-21
**Addressing Shared Concerns**

We thank all reviewers for their thoughtful and detailed feedback. We use this comment to address two shared criticisms, namely the paper’s clarity and a comparison to LMC [1].

## Clarity

We have uploaded an updated version of our paper in which we have improved the introduction and section 4 to make this paper’s technical novelties easier to distinguish. Furthermore, the new version addresses the lists of concrete suggestions provided by the reviewers. We have also clarified the experimental results in order to more clearly show the empirical evidence of our contributions.
Compared to the other scalable modular CL algorithm [3], our approach expectedly exhibits a similar performance on the sequences evaluating stability, plasticity, backward transfer and perceptual transfer. However, our method achieves superior performance on the sequences evaluating few-shot and non-perceptual transfer which our method was designed to achieve at scale. On average, we achieve +34.65 and +18.12 higher accuracy on the last problems of few-shot and non-perceptual transfer sequences, respectively.

## Comparison to LMC

Reviewers also asked for a more comprehensive comparison between our method and LMC [1].
The advantages of our approach is that it is capable of non-perceptual transfer, our approximations are more efficient, our algorithm is scalable and easier to apply to novel settings.

First, our approach combines two separate search strategies: $\\mathbb{S}^{\\text{PT}}_\\text{G}$ for perceptual and few-shot transfer and $\\mathbb{S}_\\text{BO-MAP}$ for non-perceptual transfer. LMC is similar to $\\mathbb{S}^{\\text{PT}}_\\text{G}$, but does not address non-perceptual transfer, as evident from the [1]’s results on $S^-$ in from the CTrL benchmark suite.

Similarly to $\\mathbb{S}^{\\text{PT}}_\\text{G}$, LMC computes approximations to the module’s inputs which are then used to determine each module’s relevance for processing new inputs. For this purpose LMC augments each module with a decoder which tries to reconstruct the module’s input from its output. Their approximation is computationally intensive, requiring that each new decoder’s parameters are optimized, as also roughly doubling the amount of parameters which need to be stored per module. In contrast, we use a Gaussian approximation of the distribution of a module’s projected inputs, which is cheap to compute and required orders of magnitude fewer parameters (e.g. 420 additional parameters per module in our experiments compared to 82,176 for LMC).

LMC requires that all modules are stored in memory and used in order to make a prediction. As a result, it is not a scalable algorithm, since its memory complexity is not sublinear in the number of solved problems, being $O(t)$ for $t$ solved problems. In contrast, $\mathbb{S}^{\text{PT}}_\text{G}$, is scalable since it always recommends a constant number of paths and does not require more than $1$ module per layer to be loaded at a time.

Finally, LMC has a large number of hyperparameters which makes it difficult for this algorithm to be used in new experimental settings. Concretely, applying LMC to a different modular neural architecture requires an expert to specify a separate decoder architecture for each module layer. In addition, there are more than 10 hyperparameters which guide LMC’s training process. Using LMC’s original implementation [2], we applied LMC to our new benchmark suite by adapting its decoders to the new architecture and preserving most of its HPs to values suggested by the authors. The results (Table 1) demonstrate that LMC underperforms, which we think is as a result of its hyperparameters not being carefully tuned to the new setting. We will try to further tune them and update the result if LMC’s performance improves. In contrast, our method uses the same hyperparameters for both benchmark suite, which suggests that our method is easily applicable to new experimental settings.

|                          | CCL (ours) | LMC   |
|--------------------------|------------|-------|
| $\hat{\mathcal{A}}(S^-)$       | 81.92      | 67.69 |
| $\hat{\mathcal{A}}(S^{out})$   | 78.15      | 62.77 |
| $\hat{\mathcal{A}}(S^{out*})$  | 75.72      | 60.64 |
| $\hat{\mathcal{A}}(S^{out**})$ | 75.73      | 62.64 |
| $\hat{\mathcal{A}}(S^{+})$     | 74.49      | 57.01 |
| $\hat{\mathcal{A}}(S^{in})$    | 92.82      | 69.29 |
Table 1. The average accuracy on sequences from our benchmark suite.

[1] Ostapenko, O., Rodriguez, P., Caccia, M. and Charlin, L., 2021. Continual learning via local module composition. Advances in Neural Information Processing Systems, 34, pp.30298-30312.
[2] https://github.com/oleksost/LMC
[3] Veniat, T., Denoyer, L. and Ranzato, M.A., Efficient Continual Learning with Modular Networks and Task-Driven Priors.

---

### Decision · Program_Chairs · 2023-01-20

**Decision:**

Reject

**Justification For Why Not Higher Score:**

The reviewers noted that, although the paper addresses a challenging and important problem, it has limited novelty, lacks comprehensive comparison to closely related work, and misses details on the proposed probabilistic models which makes it difficult to reproduce. The authors clarified some points in their response, but the paper would still require some more modifications to be ready for publication.

**Justification For Why Not Lower Score:**

N/A

**Metareview: Summary, Strengths And Weaknesses:**

This paper addresses the scalability issue in modular continual learning(MCL). It presents a Bayesian approach to selecting pre-trained modules based on how well the module can transfer the input.
Two different search strategies are proposed to tackle perceptual and non-perceptual transfer separately. The experiments show that the proposed framework performances comparable to state-of-the-art CL baselines.

Strengths:
- The addressed problem is challenging and important in MCL.
- The proposed method combines two search strategies to address perceptual and non-perceptual transfer differently.
- It is more efficient comparing with LMC, requires less memory and more scalable with the increase of modules, and easier to apply to new experimental settings.

Weaknesses:
- Limited novelty: The paper is built upon previous work LMC[1] which proposed using the modules' input distribution as a signal for activating the modules.
- Lack comprehensive comparison to closely related work such as Lin et al. NeurIPS 2021.
- Missing details: the probabilistic models proposed are not explained in detail.
- Limited clarity: some symbols and evaluation metrics are introduced without definition.